# Fungal parasitism on diatoms alters formation and bio–physical properties of sinking aggregates

Isabell Klawonn [1,2✉], Silke Van den Wyngaert[1,9], Morten H. Iversen [3,4], Tim J. W. Walles[1], Clara M. Flintrop[3,4,10], Carolina Cisternas-Novoa[5,11], Jens C. Nejstgaard [1], Maiko Kagami [6,7] & Hans-Peter Grossart [1,8]

Phytoplankton forms the base of aquatic food webs and element cycling in diverse aquatic systems. The fate of phytoplankton-derived organic matter, however, often remains unresolved as it is controlled by complex, interlinked remineralization and sedimentation processes. We here investigate a rarely considered control mechanism on sinking organic matter fluxes: fungal parasites infecting phytoplankton. We demonstrate that bacterial colonization is promoted 3.5-fold on fungal-infected phytoplankton cells in comparison to non-infected cells in a cultured model pathosystem (diatom *Synedra*, fungal microparasite *Zygophlyctis*, and co-growing bacteria), and even ≥17-fold in field-sampled populations (*Planktothrix*, *Synedra*, and *Fragilaria*). Additional data obtained using the *Synedra–Zygophlyctis* model system reveals that fungal infections reduce the formation of aggregates. Moreover, carbon respiration is 2-fold higher and settling velocities are 11–48% lower for similar-sized fungal-infected *vs.* non-infected aggregates. Our data imply that parasites can effectively control the fate of phytoplankton-derived organic matter on a single-cell to single-aggregate scale, potentially enhancing remineralization and reducing sedimentation in freshwater and coastal systems.

[1] Department of Plankton and Microbial Ecology, Leibniz-Institute of Freshwater Ecology and Inland Fisheries (IGB), 16775 Stechlin, Germany. [2] Department of Biological Oceanography, Leibniz Institute for Baltic Sea Research Warnemünde (IOW), 18119 Rostock, Germany. [3] Alfred Wegener Institute (AWI), Helmholtz Centre for Polar and Marine Research, 27570 Bremerhaven, Germany. [4] Centre for Marine Environmental Sciences (MARUM) and University of Bremen, 28359 Bremen, Germany. [5] Helmholtz Centre for Ocean Research (GEOMAR), 24148 Kiel, Germany. [6] Faculty of Science, Toho University, Funabashi, Chiba 274-8510, Japan. [7] Faculty of Environment and Information Sciences, Yokohama National University, Yokohama, Kanagawa 240-8502, Japan. [8] Institute of Biochemistry and Biology, Potsdam University, 14469 Potsdam, Germany. [9] Present address: Department of Biology, University of Turku, 20014 Turku, Finland. [10] Present address: The Inter-University Institute for Marine Sciences in Eilat, Eilat 8810302, Israel. [11] Present address: Ocean Sciences Centre, Memorial University of Newfoundland, St. John's, NL A1C 5S7, Canada. ✉email: isabell.klawonn@io-warnemuende.de

Phytoplankton grows fast, renewing its biomass once per week and leaving behind several gigatons of decaying organic matter in the water column of the aquatic biosphere[1,2]. Most of this decaying organic matter is remineralized within the euphotic zone, but a critical fraction gets exported to deeper layers as sinking particulate material in diverse systems, including freshwater[3,4] and coastal environments[5,6]. Particulate organic matter is thereby transported through the water column towards sediments—a mechanism that drives the bentho–pelagic coupling, i.e., the exchange of energy, mass, and nutrients between pelagic and benthic habitats. Sinking organic matter thus sustains life below the euphotic zone[7], while, on the other hand, it can also lead to expanding oxygen-deficient waters worldwide in areas where oxygen consumption through organic matter remineralization exceeds the available oxygen supply[8–10]. The fate of phytoplankton-derived organic matter is, therefore, key to aquatic food webs and biogeochemical cycles.

After blooming, phytoplankton cells together with fecal pellets and other detritus can coagulate as aggregates—referred to as lake or marine snow—which sink rapidly with velocities of up to several hundred meters per day[11]. In this regard, sinking aggregates are primary vehicles of (in-)organic matter to depth[12]. The formation, remineralization, and sedimentation of aggregates are commonly explained by physical and biological processes, the latter being intimately linked to zooplankton grazing, bacterial degradation, and viral lysis[12–16]. Recent data, however, suggests that we are still missing some crucial links in the network of biological control mechanisms on vertical organic matter fluxes[17]. Aquatic fungi, for instance, are hardly considered in this context although they are abundant and active across various aquatic systems[18,19], reaching from high-mountain lakes[20] over coastal areas[21] to the deep sea[22]. For instance, members of the fungal division Chytridiomycota, referred to as chytrids, can thrive as microparasites on phytoplankton cells. These chytrids are traditionally well documented in lakes[23], especially during bloom events when host abundances are high[24,25]. More recently, they have also gained increasing attention in coastal systems after being observed via microscopy, for instance, in the highly productive upwelling region off Chile[26], in the Arctic Ocean[27,28], and during harmful algae blooms in the Mediterranean Sea[29]. Using high-throughput sequencing methods, Chytridiomycota have also been detected in other marine regions[30–34], and DNA-based abundances of Chytridiomycota have been correlated to phytoplankton blooms or chlorophyll in coastal regions[33,35–38]. Chytrids thus occur frequently in freshwater and coastal regions where a strong benthic–pelagic coupling is commonly assumed[7,39], whereas in open ocean regions, their distribution is less evident with rare observations so far[40].

Parasitic chytrids infect up to 1–44%[24] or even 80–100% of their specific phytoplankton host population[41–43], including diatoms, dinoflagellates, and cyanobacteria. They thereby alter the dynamics of phytoplankton populations[44] and are presumed to impact organic matter remineralization and sedimentation[45]. In a natural lake, chytrid-infected phytoplankton has been shown to mainly decompose in surface waters, whereas non-infected cells settled to depth[41]. It has further been estimated that 20–25% of the photosynthetically derived carbon from the infected host population is channeled to parasitic chytrids (through the fungal shunt)[46] and further to zooplankton (through the mycoloop)[47]. Chytrids have therefore been proposed to be an integral part of aquatic food webs and element cycling[23,48]. Yet, their impact on the fate of phytoplankton biomass in terms of aggregate formation and vertical organic matter fluxes remains unknown.

Diatoms, which form fast-sinking aggregates[49,50] and often mediate high proportions of the particle export[51,52], are highly susceptible to fungal infections in freshwater and coastal systems[26,28,44]. We thus used one of the few available diatom–parasite model pathosystems (freshwater diatom *Synedra* and chytrid *Zygophlyctis*)[53] to investigate the impact of fungal infections on the formation and characteristics of sinking diatom aggregates and associated organic matter fluxes. We hereby considered cell aggregation, mass density, settling velocity, sticky polymers, bacterial colonization, and respiration—parameters that are all intimately linked to organic matter remineralization and sedimentation processes. Moreover, we complemented these culture-based investigations with analyses of bacterial abundances and infection prevalence in aggregates that were formed from field-sampled phytoplankton communities. Our results demonstrate that epidemics of fungal microparasites need to be appreciated as an episodically impactful control mechanism on organic matter fluxes in the aquatic biosphere.

## Results

Data obtained from the cultured diatom *Synedra* sp. and chytrid *Zygophlyctis planktonica* pathosystem are indicated as model system and data from the field-sampled populations as natural systems.

**Experimental setup (model system).** We incubated non-infected and chytrid-infected diatom co-cultures—hereafter non-infected and fungal-infected treatment, respectively. Both culture treatments included the freshwater pennate diatom *Synedra* and co-growing bacteria. The designated infected treatment additionally included the parasitic chytrid *Zygophlyctis*, which developed *Synedra*-associated sporangia and free-living zoospores, both being part of the life cycle of this fungal microparasite[53]. One set of both treatments (each in triplicates) was grown and sub-sampled over a growth period of nine days, to monitor the culture development (referred to as culturing). An additional set was grown for six days and thereafter transferred into rotating cylinders, to analyze the formation and characteristics of aggregates (referred to as rotating cylinders). Symbols, terms, and abbreviations used in the following are summarized in Supplementary Box S1.

**Diatom growth and infection stage (model system, culturing).** For cell enumeration, we distinguished non-infected, early-infected, maturely-infected, post-infected, and decaying *Synedra* cells (Fig. 1). Non-infected *Synedra* were recognized as healthy, intact cells with chlorophyll autofluorescence. Early-infected *Synedra* showed an encysted zoospore, which grew into a mature sporangium (incl. visible zoospores) on maturely-infected cells. Post-infected *Synedra* cells displayed remains of chitinous cell walls as a sign of previous infections after zoospore discharge. Decaying cells displayed no chlorophyll autofluorescence and also no signs of previous infections, and thus, they had most likely not undergone any fungal infection[46].

During the 9-day growth period, total *Synedra* abundances increased from approximately $0.8 \times 10^4$ to $3 \times 10^4$ cells mL$^{-1}$ (Fig. 2a, b), with similar growth rates for non-infected and infected *Synedra* populations during exponential growth (day 0–4, $0.26 \pm 0.02$ and $0.26 \pm 0.02$ d$^{-1}$, respectively, $P = 0.64$, $t$-test, $N = 3$ incubation flasks, see also Supplementary Fig. S1). In the infected treatment, the infection prevalence (including early-infected, maturely-infected, and post-infected *Synedra* cells) reached 47% on day 6 (mostly maturely-infected cells) and remained at the same percentage until day 9 (mostly post-infected cells, Fig. 2b). Time series data of chlorophyll *a* were significantly different between infected and non-infected cultures ($P < 0.0001$, generalized linear mixed model GLMM, $F_{4,16} = 217.63$), with similar chlorophyll *a* contents in both treatments during day 0–2

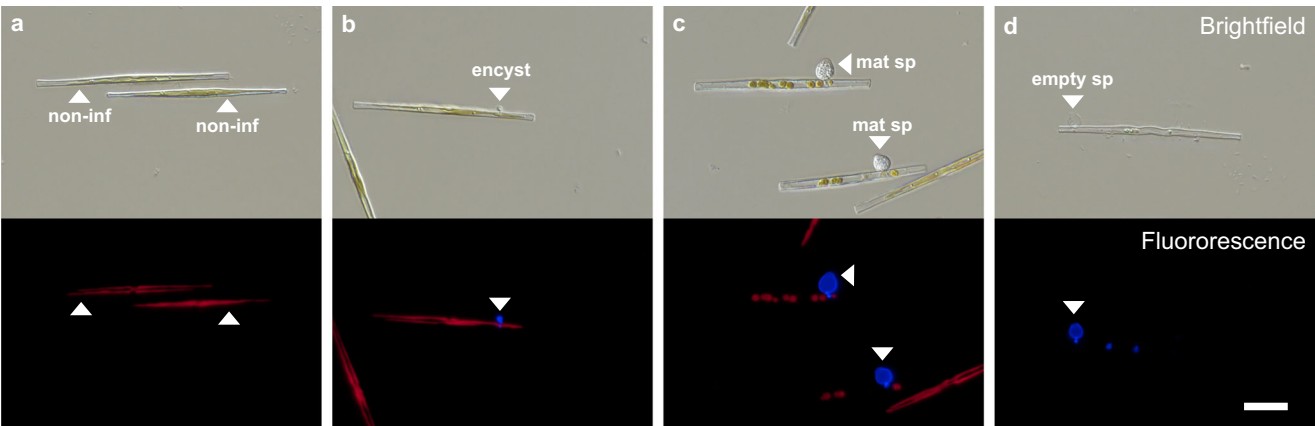

**Fig. 1 Micrographs of the diatom host (*Synedra*) and fungal microparasite (chytrid *Zygophlyctis*) at different infection stages. a** Non-infected (non-inf) healthy diatom cells. **b** Infected diatom with encysted zoospore (encyst, early-infected), which developed into a **c** mature sporangium (mat sp, maturely-infected) with new zoospores inside. The zoospores are eventually discharged into the ambient water, leaving behind an empty transparent sporangium (empty sp, post-infected) on a non-viable host cell (**d**). Shown are brightfield and epifluorescence (Fluorescence) images. Sporangia are displayed in blue (stained with Calcofluor White) and diatom chloroplasts in red (chlorophyll autofluorescence). Scale bar is 20 µm (applies to all panels).

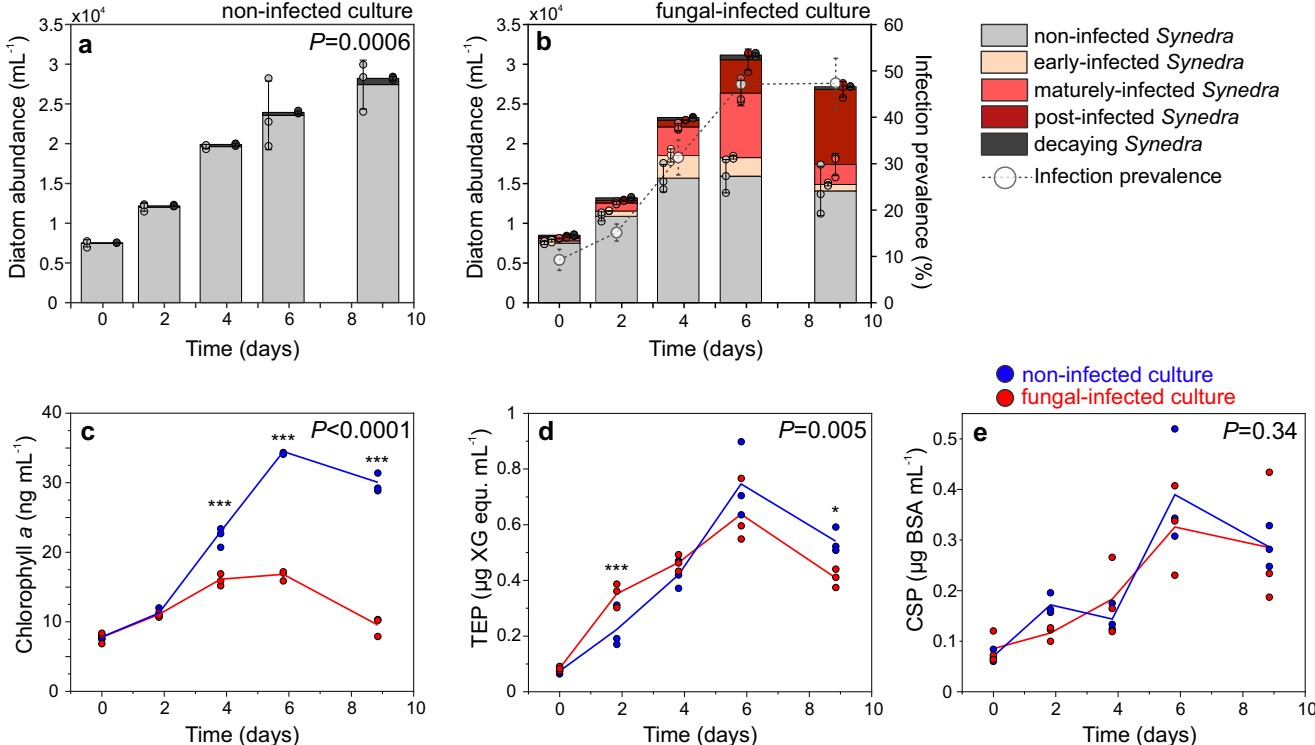

**Fig. 2 Characteristics of the diatom co-culture in the absence or presence of the fungal microparasite.** Diatom abundances and infection prevalence in the non-infected (**a**) and fungal-infected culture (**b**), **c** chlorophyll *a*, as well as **d** TEP and **e** CSP concentrations, monitored over nine days. Stacked bars **a**, **b** display mean values (bar height) with triplicate data points (circles) and standard deviations (error bars). Data in **c–e** are shown as single data points, while lines connect mean values for consecutive time points (symbols for mean values are not displayed). *P*-values, shown in the upper right corner of each graph, indicate statistically significant differences between the entire time series of each parameter in both treatments (GLMM). Statistically significant differences between treatments at single time points are indicated with asterisks (*$P < 0.05$, ***$P < 0.001$). **a–e** $N = 3$ incubation flasks. **c–e** Blue and red items represent data from the non-infected and infected cultures, respectively. TEP Transparent Exopolymer Particles, CSP Coomassie Blue Stainable Particles, XG equ. Xanthan Gum equivalents, BSA equ. Bovine Serum Albumin equivalents.

($P > 0.05$, *t*-test, $N = 3$ flasks) but significantly lower chlorophyll *a* contents in the infected *vs.* non-infected cultures during day 4–9 (*e.g.*, day 9: $9.5 \pm 1.2$ and $30.1 \pm 1.7$ ng chl *a* mL$^{-1}$, respectively, $P < 0.001$, *t*-test, $N = 3$ flasks, Fig. 2c). Nutrient concentrations (analyzed at the end of the incubations on day 9) were similar in both culture treatments ($NO_3^- + NO_2^-$: $225 \pm 2$ and $243 \pm 9$ µmol L$^{-1}$, $P = 0.07$; soluble reactive phosphorous: $31 \pm 2$ and $34 \pm 3$ µmol L$^{-1}$, $P = 0.32$; and dissolved organic carbon: $332 \pm 24$ and $299 \pm 11$ µmol L$^{-1}$ in the non-infected and infected treatment, respectively, $P = 0.12$, *t*-test, $N = 3$ flasks).

**Extracellular polymers (model system, culturing).** Transparent Exopolymer Particles (TEP) and Coomassie Blue Stainable Particles (CSP) were determined microscopically and

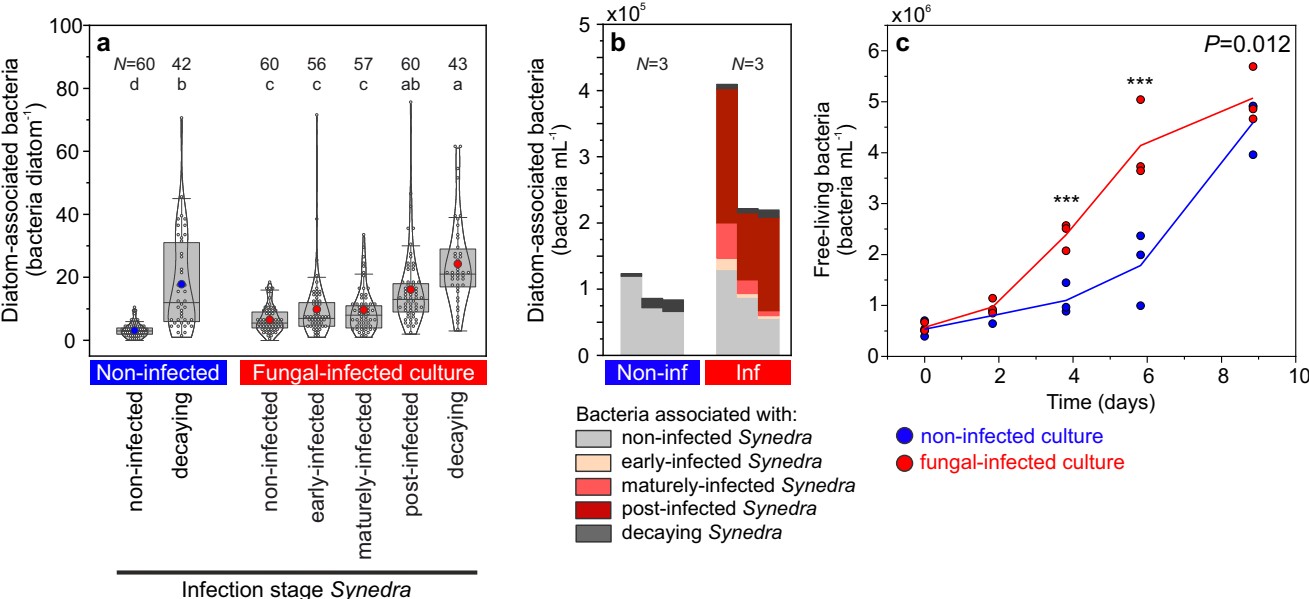

**Fig. 3 Bacterial abundances in the diatom–fungus co-cultures.** Abundances of diatom-associated bacteria are shown per individual *Synedra* cell (**a**) and per culture volume (**b**) on day 9. Bacteria were grouped based on their association with individual *Synedra* cells of different infection stages. The letters a–d in (**a**) denote significantly different groups (Kruskal–Wallis, $P < 0.05$, $N =$ numbers of analyzed *Synedra* cells). Data are shown as single data points (white-filled circles), the 25th, 50th, and 75th percentile (grey boxes), mean values (blue and red-filled circles), and distribution curves (kernel smoothing). Data from the previous sampling days 0–6 are listed in Supplementary Data 2. Stacked bars in **b** represent individual sample replicates. **c** Abundances of free-living bacteria. Single data points are plotted as circles and lines connect mean values over time (symbols of mean values are not displayed). Time series data of free-living bacterial abundances were significantly different between both treatments ($P = 0.012$, GLMM, $F_{4,16} = 4.591$). Blue and red items represent data from the non-infected and fungal-infected cultures, respectively. Statistically significant differences at single time points are indicated with asterisks (***$P < 0.001$). **b**, **c** $N = 3$ incubation flasks.

spectrophotometrically[54–56]. Photometrically-analyzed TEP concentrations increased until day 6 and decreased thereafter in both treatments (range: 0.07–0.75 µg XG equ. mL$^{-1}$, Fig. 2d). The time-dependent development of TEP concentrations, however, was significantly different between treatments ($P = 0.005$, GLMM, $F_{4,16} = 5.618$). TEP concentrations were significantly higher ($P < 0.001$) on day 2, but significantly lower on day 9 ($P = 0.04$, pair-wise comparison) in the infected treatment in comparison to the non-infected treatment. On the other sampling days, no significant differences were detected between both treatments ($P > 0.22$). Similar to TEP concentrations, CSP concentrations increased until day 6 and decreased thereafter (range: 0.07–0.39 µg BSA equ. mL$^{-1}$, Fig. 2e), but no statistically significant differences were detected between treatments ($P = 0.34$ for GLMM, $F_{4,16} = 1.2145$ and $P > 0.09$ for pair-wise comparison between single sampling days). Microscopy-derived numbers of TEP and CSP were highly variable, ranging from 119 to 3672 TEP mL$^{-1}$ ($d_{TEP} = 2$–$4\,\mu m$, $A = 3$–$12\,\mu m^2$) and from 966 to 21,196 CSP mL$^{-1}$ ($d_{CSP} = 3$–$6\,\mu m$, $A = 9$–$32\,\mu m^2$) in both culture treatments (ranges represent mean values during day 0–9, data are listed in Supplementary Data 6). No statistically significant differences in TEP and CSP sizes and areas between treatments were detected ($P > 0.13$).

**Bacterial abundance (model system, culturing).** Using fluorescence microscopy, we enumerated (i) free-living bacteria as well as diatom-associated bacteria based on their association with (ii) non-infected, (iii) early-infected, (iv) maturely-infected, (v) post-infected, and (vi) decaying *Synedra* cells. On day 9, the abundance of diatom-associated bacteria was lowest in conjunction with non-infected *Synedra* cells (6.5 ± 4.2 bacteria diatom$^{-1}$) and increased with increasing infection stage, from early-infected to

maturely-infected, post-infected, and decaying *Synedra* cells (9.9 ± 10.5, 9.6 ± 7.2, 16.1 ± 12.0, and 24.2 ± 13.7 bacteria diatom$^{-1}$, respectively, Fig. 3a). The same cell stage of *Synedra* was colonized by fewer bacteria in the non-infected culture than in the infected culture (3.1 ± 2.2 and 6.5 ± 4.2 bacteria per non-infected *Synedra* cell, respectively, and 17.8 ± 15.3 and 24.2 ± 13.7 bacteria per decaying *Synedra* cell, respectively). Consequently, diatom-associated bacteria were in total 3-times more abundant in the infected culture (Fig. 3b). Abundances of free-living bacteria increased with time, reaching significantly higher abundances on day 4 ($P = 0.0004$) and day 6 ($P = 0.0001$) in the infected culture. On day 9, however, the abundances of free-living bacteria were similar in both treatments ($P = 0.59$, t-test, $N = 3$ incubation flasks, Fig. 3c).

**Formation of sinking aggregates (model system, rotating cylinders).** Aggregate formation was reduced substantially in the presence of the fungal parasite, that is, aggregates were less abundant and smaller in fungal-infected *vs.* non-infected *Synedra* treatments (Fig. 4a–d). In detail, during the initial 12 h of aggregate formation, small aggregates ($d = 0.4$–$1.3\,mm$) were 38 ± 14-times less abundant (range: 20–58-times, # L$^{-1}$) and comprised 46 ± 24-times less cumulative aggregate volume (range: 13–78-times, mm$^3$ L$^{-1}$, $N = 5$ time points) in the fungal-infected *Synedra* treatment. Thereafter, also large aggregates (>1.3–5.6 mm) became abundant, and both size classes ($d = 0.4$–$1.3\,mm$ and >1.3–5.6 mm) were 6 ± 4-times less abundant (range: 2–14-times) and comprised 9 ± 7-times less cumulative aggregate volume (range: 3–25-times, $N = 10$ time points) during fungal infections. Aggregates within the smallest size bin ($d = 0.35$–$0.46\,mm$) were most abundant (up to 80 and 26 aggregates L$^{-1}$ in the non-infected and fungal-infected treatment,

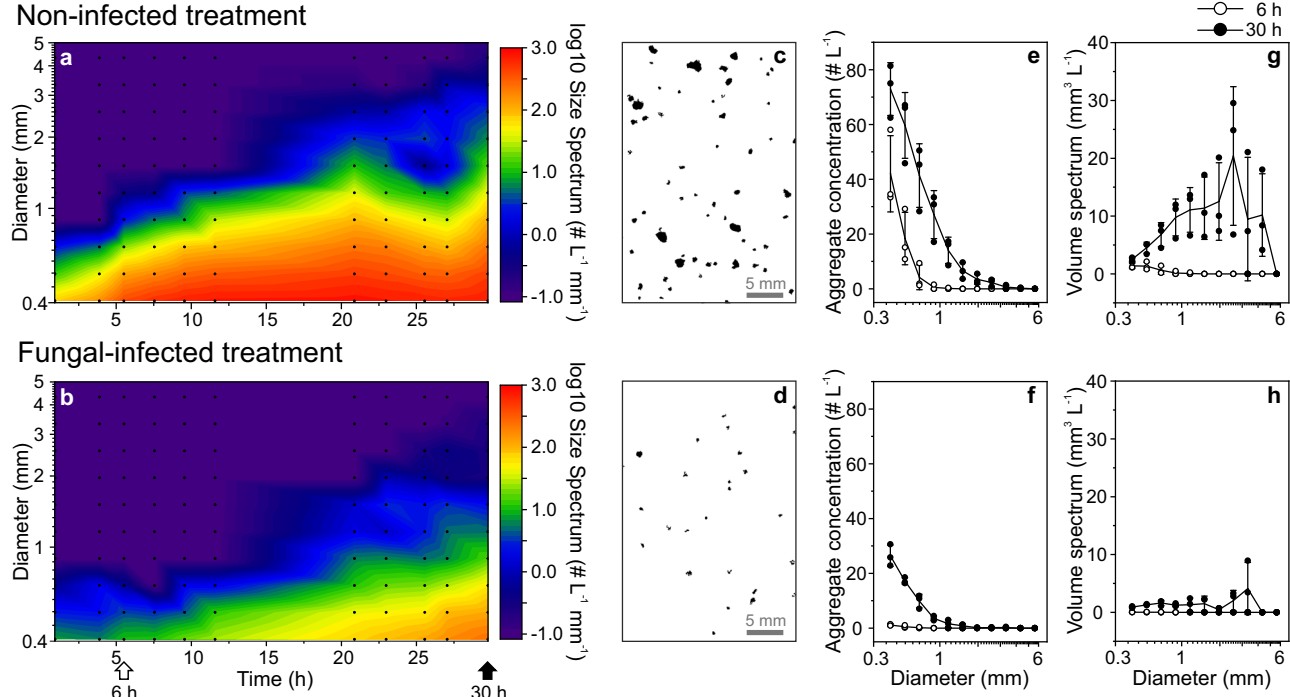

**Fig. 4 Aggregate formation from non-infected (upper panel) and fungal-infected (lower panel) diatom cultures, recorded in rotating cylinders via imaging. a, b** Size spectra of aggregates during the initial 30 h of cylinder rotation. Size spectra show the number of aggregates per water volume normalized to the width of each size bin. Log-transformation was used to visualize the wide size spectrum range. Values of −1.0 indicate that no aggregates in the respective size bin were present (*i.e.*, $\log_{10}(0)$ was set to −1.0). Small black dots mark the sampled time points and size bins. **c, d** Exemplary particle images recorded after 30 h. **e–h** Aggregate concentration and volume spectra for different size bins are shown for aggregates imaged after 6 h and 30 h. The diameter represents the equivalent circular diameter (*ECD*). Data in **e–h** are shown as single data points (triplicates) and lines connect mean values for consecutive time points (symbols for mean values are not displayed, but error bars represent standard deviations). Empty and filled circles show data after 6 and 30 h. The imaging setup is shown in Supplementary Fig. S2. Supplementary Fig. S3 shows exemplary aggregates imaged after sampling and the aggregation status after 45 h in the infected treatment.

respectively, Fig. 4e, f), while large aggregates ($d = 2.2–3.8$ mm) accounted for most of the cumulative aggregate volume (up to 20 and 2 $mm^3 L^{-1}$ in the non-infected and fungal-infected treatments, respectively, Fig. 4g, h).

**Bacterial abundance and infection prevalence in aggregates and the ambient water (model system, after cylinder rotation).** We sampled individual aggregates from the rotating cylinders to analyze their bio–physical properties. Those aggregates were similar in size, with mean diameters of $2.4 \pm 1.3$ mm for non-infected aggregates and $2.4 \pm 1.0$ mm for infected aggregates ($P = 0.93$, *t*-test, Table 1). Bacteria were 5-times more abundant on fungal-infected aggregates than on non-infected aggregates ($12.3 \pm 0.8 \times 10^4$ and $2.7 \pm 0.5 \times 10^4$ bacteria $agg^{-1}$, $P < 0.0001$, *t*-test, Fig. 5a) due to two aspects. Firstly, abundances of diatom-associated bacteria were lowest on non-infected *Synedra*, and increased consecutively on early-infected, maturely-infected, post-infected, and decaying *Synedra*, (0.8–18.8 bacteria $diatom^{-1}$, Fig. 5b). And secondly, fungal-infected cells were 1.7-times enriched within aggregates as compared to the ambient water ($71 \pm 3\%$ and $42 \pm 3\%$ infection prevalence, respectively, $P < 0.0001$, *t*-test, Fig. 5c). This enrichment was mostly due to post-infected cells, which were 2.6-times more abundant within the aggregates as compared to the ambient water ($51 \pm 6\%$ and $19 \pm 3\%$ relative abundance, $P < 0.0001$, *t*-test, Fig. 5d).

**Physical parameters and respiration of individual aggregates (model system, after cylinder rotation).** The mass density $\rho_{s-agg}$

of aggregates, *i.e.*, the density of the particulate material in hydrated aggregates (also termed solid hydrated density)[57], was significantly higher for non-infected aggregates than for infected aggregates ($1.284 \pm 0.003$ and $1.258 \pm 0.004$ g $cm^{-3}$, respectively, $P < 0.0001$, *t*-test, Fig. 5e, Table 1). Similarly, mass densities of single cells $\rho_{s-cells}$ were significantly higher for non-infected *Synedra* cells than for infected *Synedra* cells ($1.269 \pm 0.001$ and $1.251 \pm 0.004$ g $cm^{-3}$, respectively, $P = 0.002$, *t*-test, Table 1). Excess densities, defined as the mass densities of aggregates in excess relative to the ambient water, were also significantly higher for non-infected aggregates in comparison to fungal-infected aggregates ($0.8 \pm 0.3$ and $0.6 \pm 0.1$ mg $cm^{-3}$, Welch-test, $P = 0.009$, see Supplementary Fig. S4a for the size~excess density curve fit). Settling velocities ranged from 52–534 m $d^{-1}$, and increased with increasing aggregate size. The size~settling velocity curve fit (power function as used by[58]) was significantly different between non-infected and infected aggregates ($P = 0.04$, *F*-test comparing the curve fit, $F_{1,22} = 3.651$, Fig. 5f). The fractal dimension $D_3$, which describes the fractal geometry of aggregates, was significantly lower for non-infected aggregates than for fungal-infected aggregates ($2.37 \pm 0.30$ and $2.64 \pm 0.24$, $P = 0.005$, Table 1), indicating a more compact, less porous structure of infected aggregates (see Supplementary Fig. S4b for the curve fit used to derive $D_3$). This indication, however, was not confirmed by similar porosities of non-infected and infected aggregates (Table 1) and faster settling velocities for non-infected *vs.* infected aggregates (Fig. 5f). Carbon contents were similar for both aggregate types of similar sizes ($6.7 \pm 2.8$ and $8.0 \pm 5.3$ μg POC $agg^{-1}$, $P = 0.52$, *t*-test), but carbon-specific respiration rates were

**Table 1 Characteristics of single aggregates formed from non-infected and fungal-infected diatom cultures.**

| Parameter | Non-infected aggregate | N | Fungal-infected aggregate | N | Unit | P-value | Method/Equation |
|---|---|---|---|---|---|---|---|
| *Set I* | | | | | | | |
| Diameter $d$ | 2.4 ± 1.3 (1.1–4.9) | 12 | 2.4 ± 1.0 (1.3–4.7) | 12 | mm | 0.93 | Image analyses |
| Porosity $\varphi$ | 0.9993 ± 0.0002 (0.9990–0.9997) | 12 | 0.9995 ± 0.0002 (0.9993–0.9997) | 12 | - | 0.56 | Eq.4 |
| Excess density $\Delta\rho$ | 0.8 ± 0.3 (0.4–1.3) | 12 | 0.6 ± 0.1 (0.4–0.9) | 12 | mg cm$^{-3}$ | **0.009** | Eq.5 |
| Drag coefficient $C_D$ | 13.1 ± 11.4 (2.2–43.2) | 12 | 14.9 ± 9.8 (2.9–36.9) | 12 | - | 0.69 | Eq.6 |
| Reynolds number $Re$ | 6.8 ± 8.7 (0.6–28.2) | 12 | 4.6 ± 5.5 (0.7–18.1) | 12 | - | 0.46 | Eq.7 |
| Settling velocity $U$ | 201 ± 148 (52–534) | 12 | 144 ± 103 (52–363) | 12 | m d$^{-1}$ | **0.04** (curve fit) | Settling column |
| Fractal dimension $D_3$ | 2.37 ± 0.30 | 12 | 2.64 ± 0.24 | 12 | - | **0.005** (curve fit) | Eq.8 |
| *Set II* | | | | | | | |
| POC content[a] | 6.7 ± 2.8 (3.6–11.4) | 9 | 8.0 ± 5.3 (1.6–15.4) | 9 | µg C agg$^{-1}$ | 0.52 | Carbon analyzer |
| C-specific respiration | 0.16 ± 0.08 (0.08–0.31) | 9 | 0.34 ± 0.08 (0.22–0.45) | 9 | d$^{-1}$ | **0.0002** | Oxygen sensor |
| *Set III* | | | | | | | |
| Bacterial colonization[b] | 2.7 ± 0.5 × 10$^4$ (1.8–3.3 × 10$^4$) | 9 | 12.3 ± 0.8 × 10$^4$ (11.0–13.3 × 10$^4$) | 9 | agg$^{-1}$ | **<0.0001** | Microscopy |
| *Set IV* | | | | | | | |
| Mass density aggregates $\rho s\text{-}agg$ | 1.284 ± 0.003 (1.280–1.289) | 6 | 1.258 ± 0.004 (1.253–1.262) | 4 | g cm$^{-3}$ | **<0.0001** | Density gradient |

| Set V (Fungal-infected treatment) | Ambient water | | Aggregate | | | | |
|---|---|---|---|---|---|---|---|
| Infection prevalence | 42 ± 3 (37–45) | 6 (1 mL) | 71 ± 3 (66–76) | 9 | % | **0.0001** | Microscopy |

| Co-cultures (Single Synedra cells[c]) | Non-infected cells | | Infected cells (incl. sporangium) | | | | |
|---|---|---|---|---|---|---|---|
| Mass density cells $\rho s\text{-}cells$ (single diatom cells) | 1.269 ± 0.001[d] (1.268–1.271) | 3 (flasks) | 1.251 ± 0.004 (1.249–1.256) | 3 (flasks) | g cm$^{-3}$ | **0.008** | Density gradient |
| Cell biovolume | 1446 ± 265 (855–2161) | 50 (cells) | 1688 ± 238 (1217–2701) | 20 (cells) | µm$^{-3}$ | **0.005** | Microscopy |

The similar-sized aggregates were picked individually after their formation in rotating cylinders. Aggregates were grouped into five sets (set I–V, each with 4–12 replicates) since not all parameters could be measured on the same aggregate. N denotes the number of aggregates, if not specified differently. Bold P-values indicate statistically significant differences between the non-infected and fungal-infected treatments.
[a]POC – particulate organic carbon.
[b]assuming 20,000 diatom cells per aggregate since we counted 22,843 ± 34,443 *Synedra* cells per aggregate ($d = 2.1 ± 0.8$ mm in set III, $N = 18$).
[c]sampled directly from the co-cultures (not from the rotating cylinders).
[d]mass density is given for non-infected *Synedra* cells that were sampled from the fungal-infected treatment, while it was similar for non-infected *Synedra* cells in the non-infected treatment (1.270 ± 0.006 g cm$^{-3}$, range 1.264–1.276 g cm$^{-3}$, $P = 0.97$, $N = 3$).

on average 2-fold higher for fungal-infected aggregates as compared to non-infected aggregates (0.16 ± 0.08 and 0.34 ± 0.08 d$^{-1}$, $P = 0.0002$, *t*-test, Fig. 5g). These respiration rates are high compared to previously measured rates of laboratory-formed and field-sampled aggregates (approx. range: 0.06–0.13 d$^{-1}$)[6,57,59–63]. Due to the slower settling velocities and higher respiration rates, infected aggregates were estimated to respire 2.4–3.7-times more carbon per meter settled than non-infected aggregates (diameter range: 1–5 mm, Fig. 5h).

**Bacterial abundance and infection prevalence of aggregates formed from field-collected plankton communities (natural systems, incl. cylinder rotation).** We sampled a natural phytoplankton community in a temperate lake (Lake Stechlin), which comprised mostly filamentous cyanobacteria (*Planktothrix*, 398 ± 74 filaments mL$^{-1}$ and *Aphanizomenon/Pseudanabaena*, combined 443 ± 11 filaments mL$^{-1}$) and diatoms (*Synedra* 280 ± 63 cell mL$^{-1}$ and *Stephanodiscus/Fragilaria*, combined

1382 ± 345 cells mL$^{-1}$). Chytrid infections were highest on *Planktothrix*, *Synedra*, and *Fragilaria* cells, and thus, we inspected these taxa in more detail (~2% infection prevalence during our sampling, while prevalences of up to 44% on *Synedra* and *Fragilaria* have been observed previously at the same location)[24]. Bacterial abundances were ≥17-times higher on infected cells than on non-infected cells (*Planktothrix* 0.1 ± 1.2 and 13.4 ± 14.8 bacteria 100 µm-filament$^{-1}$, *Synedra*: 0.9 ± 2.2 and 17.5 ± 20.9 bacteria diatom$^{-1}$, and *Fragilaria*: 0.5 ± 1.5 and 8.8 ± 6.3 bacteria diatom$^{-1}$, respectively, Fig. 6a). For *Planktothrix* and *Fragilaria*, fungal-infected cells were relatively more abundant in aggregates than in the ambient water (*Planktothrix*: 3.4 ± 2.7% and 0.2 ± 0.4% infection prevalence, $P < 0.001$, *t*-test, $N = 30$ and 29, respectively; and *Fragilaria*: 2.1 ± 1.6% and 0.5 ± 1.3%, $P < 0.001$, *t*-test, $N = 30$ and 29, respectively, Fig. 6b). In contrast, relative abundances of infected *Synedra* cells were similar in both aggregates and the ambient water (2.6 ± 3.1% and 2.7 ± 3.5% prevalence, $P = 0.47$, *t*-test, $N = 30$ and 29, respectively).

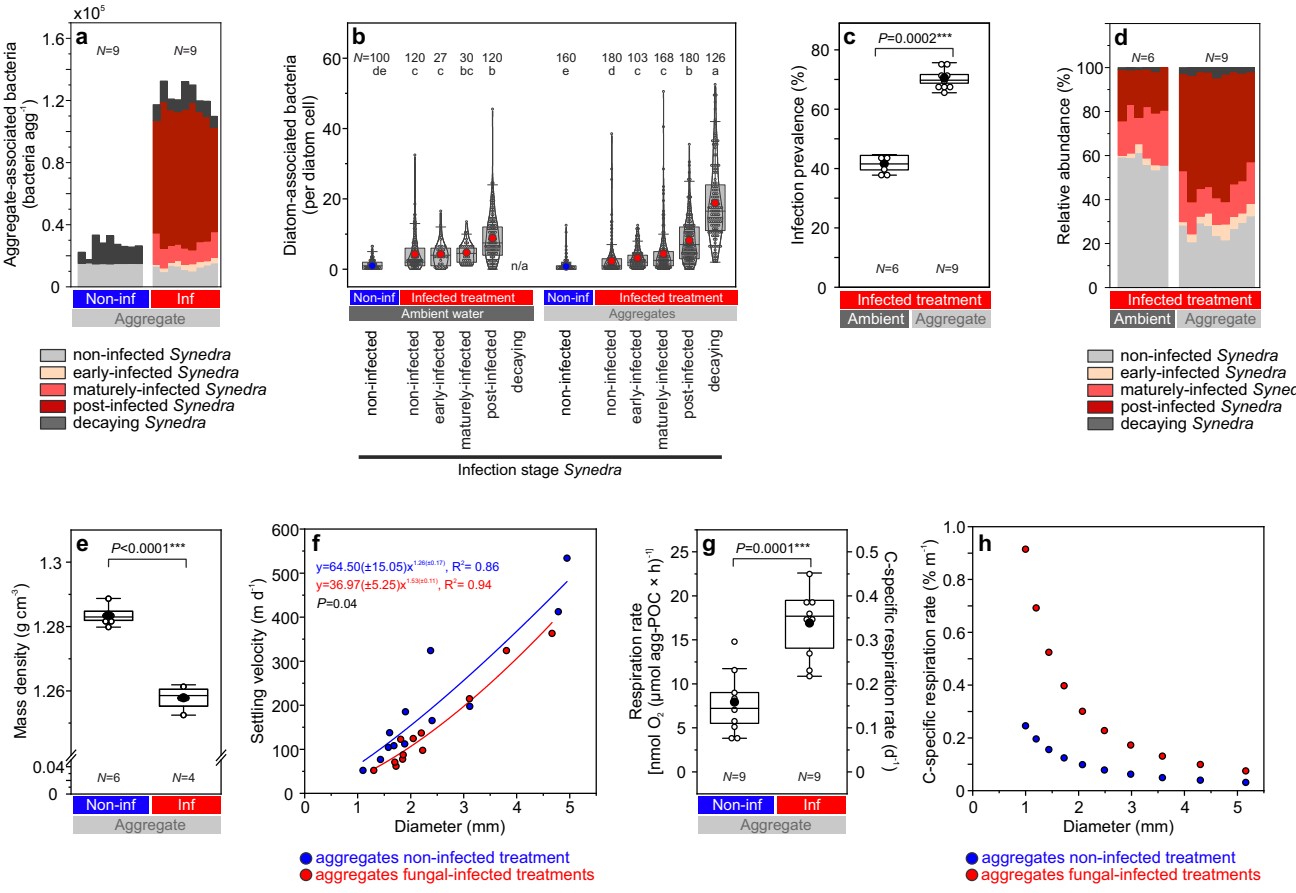

**Fig. 5 Bio-physical properties of single aggregates formed from non-infected and fungal-infected diatom cultures in rotating cylinders.** Bacterial abundances on whole aggregates (**a**) and single *Synedra* cells (**b**), infection prevalence (**c**), and relative abundances of different *Synedra* cell types within aggregates and in the ambient water (**d**). Bacteria in **a** were grouped based on their association with individual *Synedra* cell types within single aggregates. The letters a–e in **b** denote significantly different groups (Kruskal–Wallis, $P < 0.05$). *N* indicates the number of analyzed cells. Data in **b** are shown as single data points (white circles), the 25th, 50th, and 75th percentile (grey boxes), mean values (blue and red-filled circles), and distribution curves (kernel smoothing). Stacked bars in **a**, **d** represent individual replicates. Mass density (**e**), settling velocities (**f**), and respiration rates (**g**, **h**) of single aggregates. **f** Settling velocities were positively correlated with the aggregate diameter. Fitted curves (power function) were significantly different between both treatments ($P = 0.04$). Blue and red items represent data from the non-infected and fungal-infected aggregates, respectively. **g** Respiration rates given as nmol $O_2$ per µmol agg-POC × h define the oxygen consumption per mol aggregate-POC and hour. Rates of 0.1 d$^{-1}$ indicate that 10% of the aggregate-POC were respired per day. **h** Respiration rates per meter settled, estimated for differently-sized aggregates. Rates of 0.1% m$^{-1}$ indicate that 0.1% of the aggregate-POC were respired per meter settled. Data in **c**, **e**, **g** are shown as single data points (white circles), the 25th, 50th, and 75th percentile (white boxes), and mean values (black circles).

## Discussion

Understanding the sinking flux of particulate organic matter marks a long-standing frontier in aquatic science, while over the last two decades, the consideration of the multiple mechanisms that control those fluxes has expanded substantially[14,64–67]. We suggest that epidemics of eukaryotic microparasites, such as the herein-studied parasitic fungi, should be regarded as an additional biological mechanism that can attenuate vertical organic matter fluxes, next to the commonly considered bacterial degradation, viral lysis, and zooplankton grazing[64,68,69]. We build upon this suggestion in the following discussion. Our data, however, derive mostly from one model system and from the freshwater environment. We, therefore, encourage future studies on eukaryotic parasites in diverse aquatic systems to uncover their full potential to modulate sinking organic matter fluxes. Moreover, parasitic infections induce cascading effects on phytoplankton community composition[70], zooplankton production[71,72], and carbon recycling[46], all of which are known to alter vertical mass fluxes[6,50,73–77]. The weighted impact of parasite epidemics on sinking organic matter fluxes thus likely varies in multifarious plankton networks—variations that we are only starting to disentangle.

The photosynthetic biomass of the diatom host population was substantially diminished during fungal infections, as implied by up to 3-fold lower chlorophyll *a* at 47% infection prevalence in comparison to the non-infected culture (Fig. 2c). Whether this decrease in chlorophyll results in an actual decrease in photosynthetic activity has not been quantified to date, but in seaweeds, chytrid infections were shown to reduce the photosynthetic efficiency[78]. The reduction in photosynthetic pigments may thus also reduce the photosynthetic biomass that could get exported to depth. Bacterial abundances, especially those of phytoplankton-associated bacteria, were promoted up to 3-fold on the cultured *Synedra* cells during fungal infections (Fig. 3a, b) in line with previous laboratory-based studies[46,71,79,80]. We here confirmed this pattern also for field-sampled populations, which demonstrated an even higher—at least one order of magnitude—increase in bacterial colonization on fungal-infected *vs.* non-infected phytoplankton cells (Fig. 6a). This observation may be explained by

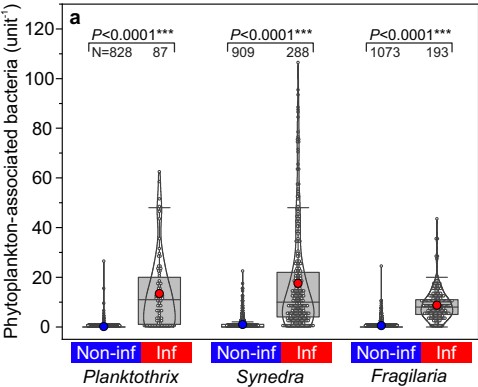

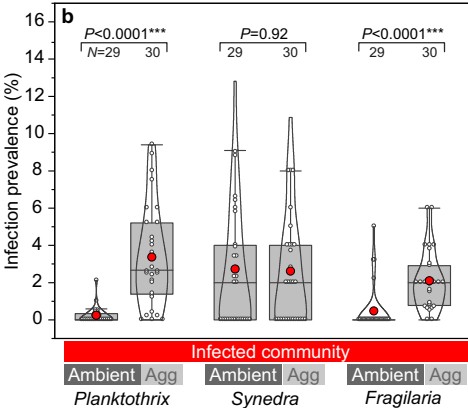

**Fig. 6 Bacterial abundances and infection prevalence in field-sampled phytoplankton. a** Bacterial abundances on non-infected and fungal-infected phytoplankton cells. The numbers of associated bacteria are given per 100 μm filament length (*Planktothrix*) or per diatom cell (*Synedra* and *Fragilaria*). **b** Infection prevalence in the ambient water and aggregates, after cell aggregation in rotating cylinders. *N* indicates the number of analyzed cells **a** or the number of aggregates and ambient water samples **b**. Data are shown as single data points (white circles), the 25th, 50th, and 75th percentile (grey boxes), mean values (blue and red circles), and distribution curves (kernel smoothing).

phytoplankton–bacteria interactions that are more essential for cell growth under nutrient-depleted conditions in nature, as opposed to nutrient-replete conditions in culture. Bacterial colonization on infected phytoplankton was possibly promoted by bacterial chemotaxis as bacteria are commonly attracted by decomposing, nutrient-leaking phytoplankton cells[81–83]. Additionally, healthy diatom cells can actively repel opportunistic bacteria through chemical signaling[84], an ability that fungal-infected host cells might have lost and consequently have become overgrown by bacteria in our incubations.

Parasitic infections reduced the formation of sinking diatom aggregates substantially. For instance, aggregates larger than 0.5 mm, which can dominate the downward flux in natural systems[85,86], reached abundances of $5 \, L^{-1}$ after 6 h when parasitic fungi were absent, whereas it took 12 h to reach comparable abundances when the parasites were present (Fig. 4). Aggregate formation in rotating cylinders generally depends on cell concentrations and cell-to-cell stickiness[12,13]. In our culture incubations, cell concentrations were similar in both treatments (~5000 *Synedra* $L^{-1}$, resembling *Synedra* blooms[87]), but the stickiness, *i.e.*, the probability that two cells coagulate upon encounter, was presumably reduced during fungal infections. The stickiness of cells and particles can be increased by the presence of

exopolymers like TEP and CSP, which are broadly classified as acidic polysaccharides and alkaline amino acids, respectively[88,89]. TEP concentrations showed significantly different dynamics over time in our non-infected *vs.* infected *Synedra* culture (Fig. 2d). However, differences in absolute TEP concentrations on each sampling day were minor, and we presume that fungal infections changed the molecular composition rather than concentration of TEP. Concentrations of CSP were similar in both culture treatments, supporting previous suggestions that CSP are less involved in aggregate formation than TEP[90].

Interestingly, infected cells preferentially aggregated over non-infected cells in the fungal-infected population, and yet, overall aggregation was reduced in comparison to non-infected populations. To explain this observation, we consider mechanical and chemical stickiness[91]. Concerning mechanical stickiness, the development of fungal sporangia on the outside of infected *Synedra* cells increased the effective size of cells, potentially increasing the interparticle encounter rate and entangling of cells (see Supplementary Fig. S5 for illustration). Concerning chemical stickiness, we may need to distinguish between cell-associated and freely-suspended polymers, as shown recently for diatoms[92]. We suggest that fungal infections shifted polysaccharides towards (1) fewer freely-suspended polymers, reducing the overall aggregation, but (2) more adhesive polymers at the cell surface of infected *Synedra* cells, enhancing their cell-to-cell stickiness. Indeed, the observed faster growth of bacterial populations in the infected treatments (Fig. 3) indicates a modification in the organic carbon pool during fungal infections, as shown previously[79]. Moreover, bacterial activities, which are impacted by fungal infections[46], are known to either promote or inhibit aggregate formation[93,94]. We, therefore, hypothesize that freely-suspended polymers were degraded more effectively by free-living bacteria during fungal infections, whereas cell-associated polymers accumulated at the surface of post-infected diatoms. To test this hypothesis, we recommend investigating the molecular composition and stickiness of both cell-associated and freely-suspended polymers during parasitic infections.

*Synedra* cells in culture, as well as field-sampled *Planktothrix* and *Fragilaria*, aggregated preferentially when being infected as compared to their non-infected counterparts. In contrast, infected *Synedra* cells sampled in the field did not show such preferential aggregation since non-infected and infected cells were proportionally equally distributed in aggregates and the ambient water. Our *Synedra* cell isolate originated from a different lake (Lake Haussee) than the field-sampled cells (Lake Stechlin), and their morphologies were different (*e.g.*, field-sampled cells were ca. 4-times longer than the cell isolate). The different aggregation patterns between cultured and field-sampled *Synedra* may thus have resulted from genera-specific differences (involving diatom-specific bacteria interactions) or even strain-specific differences, as known from viral infections on phytoplankton[95]. Moreover, in culture, post-infected *Synedra* cells showed a higher aggregation potential than early-infected and maturely-infected cells (Fig. 5). The infection stage is therefore crucial when examining aggregation dynamics. In the field-sampled populations, we did not differentiate between different infection stages (cells were solely classified as non-infected or infected), but considering the low infection prevalence, we suggest that the field-sampled *Synedra* population was at an early epidemic stage with mostly early-infected cells.

The development of sporangia on the outside of the host cell expanded the biovolume of diatom cells by 16 ± 5% (*N* = 20 cells, Table 1) with mostly organic material with, according to literature values, rather low density (cytoplasm: $1.10 \, g \, cm^{-3}$, chitin: $1.425 \, g \, cm^{-3}$) as compared to siliceous frustules ($1.82 \, g \, cm^{-3}$)[96,97]. The lower mass density of infected *Synedra* cells and aggregates in

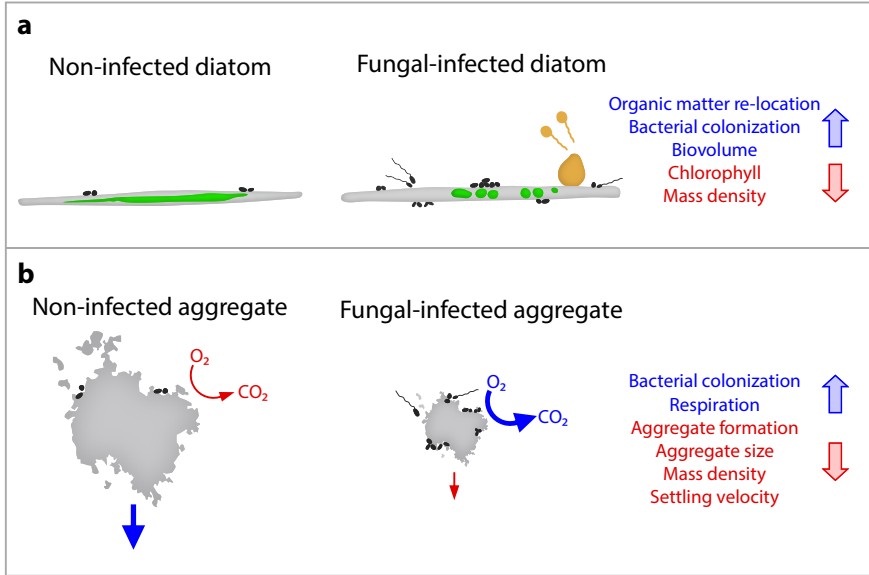

**Fig. 7 Illustration of the impact of fungal parasites on biological and physical properties of single diatom cells and aggregates.** The blue and red text and arrows indicate the fungal-induced increase and decrease of each property, respectively. Diatom cells **a** and aggregates **b** are illustrated in light grey, diatom chlorophyll in green, fungal sporangium and zoospores in orange, and bacteria in dark grey.

comparison to non-infected cells and aggregates may further be explained by more cell-associated polymers (such as TEP), fewer cells per aggregate, or thinner silica frustules as bacteria are known to dissolve silica[98]. Following Stokes' law (Eq. 5), lower mass densities result in slower settling, which was confirmed by two lines of evidence in our data set. Firstly, based on the size~settling velocity regression (Fig. 5f), we estimated that infected aggregates (71% prevalence, $d = 1–5$ mm) sank 11–48% slower than non-infected aggregates. And secondly, the herein-measured lower mass densities (measured in a density gradient) and lower excess densities (Eq. 4) of infected *vs.* non-infected aggregates would both result in ca. 30% slower settling velocities of infected aggregates (following Stokes' law), similar to the size~settling velocity regression. We thus conclude that host-associated sporangia increased the cell size of diatom cells but reduced their mass per volume (g cm$^{-3}$), slowing down their settling. Reduced aggregate formation and hence, smaller aggregates during fungal epidemics may have an even stronger impact on sinking velocities. For example, maximum aggregate diameters were 1 mm in the non-infected treatment but only 0.5 mm in the infected treatment after 6 h of aggregate formation. Such a 2-fold decrease in aggregate diameter led to 3-fold slower sinking velocities (according to the size~settling velocity regression in Fig. 5f).

Aggregates that comprised 71% fungal-infected cells respired twice as much of their carbon content per day as compared to non-infected aggregates with similar size and POC content (Fig. 5g). We presume that those higher respiration rates were due to the above-mentioned increase in bacterial abundances during fungal infections and due to the respiration of the para-sitizing fungi. Chytrid fungi grow as host-associated sporangia, which after maturation, discharge free-swimming zoospores to seek out a new host. Especially those zoospores are expected to have a high oxygen demand since they are rich in sterols and fatty acids[99,100] to support their intensive swimming and seeking behavior. Considering the herein determined carbon respiration rates and settling velocities (Fig. 5h), we roughly estimated that infected aggregates ($d = 1$ mm) would have lost 37% and non-infected aggregates only 12% (*i.e.*, 3-times less) of their initial carbon content after settling 50 m, indicating a substantial reduction in export efficiencies due to fungal infections.

Taken together, our incubations revealed that fungal infections decreased the chlorophyll content, cell mass density, cell aggrega-tion, aggregate size, and settling velocity, but increased bacterial colonization and carbon respiration on a single-cell to single-aggregate scale (Fig. 7). As a result, fungal infections may be expected to shift organic matter fluxes towards decreased sedi-mentation and increased remineralization through several mechanisms, with implications for ecosystem-wide bentho–pelagic coupling. Given that fungal parasites are widespread in aquatic systems across different climate zones globally[27–29,44], including productive upwelling regions[26], commercial mass cultures[101], and areas impacted by harmful algal blooms[29], their epidemics thus have the potential to diminish the strength and efficiency of vertical organic matter fluxes in natural and engineered aquatic environments.

## Materials and methods

**Experimental design (model system)**. The model pathosystem comprised the pennate diatom host *Synedra* sp. Ehrenberg, 1830 (also referred to as *Ulnaria*[102], strain HS-SYN2, $l = 87.6 \pm 4.1$, $w = 3.3 \pm 0.5$, and $h = 5.0 \pm 0.8$ μm, $N = 50$ cells) and the parasitic chytrid *Zygophlyctis planktonicum* (strain SVdW-SYN-CHY1), includ-ing host-associated sporangia (7.5 $\pm$ 0.8 μm, $N = 20$) and zoospores ($d = 3$ μm, measured in[53]) isolated from lakes in Northern Germany[53]. Bacteria were co-isolated with the diatom and chytrid and maintained in co-culture in a nutrient-replete medium for several months, which likely selected for copiotrophic taxa[46]. Chytrid infections on the diatom host were initiated by free-swimming zoospores, which attached to a host cell, encysted, and developed into epibiotic sporangia while penetrating and digesting the host's interior through a rhizoidal system[53]. Within the (zoo-)sporangium, the next generation of zoospores was formed and finally released through the rupture of the sporangium. One infection cycle took 1–2 days. Each infection was lethal and prohibited further reproduction of the host.

Batch cultures were grown in CHU-10 medium (Supplementary Table S1) at constant temperature (17 °C). The light regime was 16:8 h, providing 40 μE s$^{-1}$m$^{-2}$ during the 16-h light phase. *Synedra* was grown as $12 \times 650$ mL in 1 L Erlenmeyer flasks until reaching ca. 7500 cells mL$^{-1}$. Half of those flasks ($6 \times 650$ mL) were thereafter inoculated with a *Synedra–Zygophlyctis* co-culture (40 mL with ca. 98% prevalence), including mature fungal sporangia, which were projected to discharge new zoospores within the next hours (infected treatment). The resulting infection prevalence was 9% in the infected treatment on day 0. The other $6 \times 650$ mL flasks remained without *Zygophlyctis* (non-infected treatment). To monitor the culture development over time, $6 \times 650$ mL (three per treatment) were grown and sub-sampled for nine days (see culture characteristics). The remaining $6 \times 650$ mL (three per treatment) were grown for six days and subsequently transferred into rotating cylinders to follow the formation of aggregates (see formation and

characterization of sinking aggregates). Erlenmeyer flasks were gently shaken by hand once per day for cell resuspension and water mixing.

**Culture characteristics (model system, culturing).** Sub-samples were taken from the non-infected and fungal-infected diatom cultures (in triplicates) after 0, 2, 4, 6, and 9 days. CHU-10 medium served as a blank for all analyses.

For chlorophyll $a$ and nutrients analyses, 3 mL were transferred into 15 mL Falcon tubes and centrifuged at 3000 rpm for 20 min at 4 °C. The supernatant was gently removed and the remaining pellet was extracted in 90% acetone, frozen at −20 °C, and analyzed after one month using a fluorometer (436/680 nm, Hitachi Fluorescence Spectrometer F-7000). Concentrations of chlorophyll $a$ were calculated and corrected for phaeopigments, measured after acidification[103]. Calibration was done with a chlorophyll $a$ standard (*Anacystis nidulans*, Sigma C6144, 0.3–300 ng mL$^{-1}$, precision ±1%). Concentrations of nitrate/nitrite and soluble reactive phosphorous were determined from 0.45 μm-filtered water by flow injection analysis (FIA) and spectrometric detection (ISO-13395-D28 and ISO/DIS-15681-2, respectively). Concentrations of dissolved organic carbon (DOC) were analyzed from GF/75-filtered water on a TOC-V CPH with a nondispersive infrared sensor (Shimadzu, Kyoto, Japan) using combustion catalytic oxidation.

Diatom abundances and infection prevalence were analyzed from 2 mL sub-samples, which were transferred into microcentrifuge tubes, preserved with Lugol (50 μL per 100 mL sample, see Supplementary Table S1 for Lugol recipe), and stored at 4 °C in the darkness. *Synedra* cells were counted in Utermöhl plankton chambers (Hydrobios, Germany) under an inverted epifluorescence microscope (Nikon Ti2-E, x400 magnification). The chitinous cell walls of sporangia were stained with Calcofluor White (CFW, 5 μg mL$^{-1}$, excitation wavelength Ex 387/11, emission wavelength Em 442/46 nm, Merck F3543)[104]. *Synedra* cells were differentiated as (i) non-infected, (ii) early-infected, (iii) maturely-infected, (iv) post-infected, and (v) decaying cells (see Fig. 1 for cell type i–iv). The chlorophyll autofluorescence was inspected at Ex 635/18 nm and Em 680/42 nm. We counted at least 50 cells per group (i–iv), or half of the chamber if fewer cells were present.

For bacterial abundance analyses, 0.5–1 mL were preserved with paraformaldehyde (PFA, 1.5% final concentration), filtered onto polycarbonate filters (PC, 0.2 μm, 25 mm, Whatman), and stored at −20 °C. Prior analyses, filters were stained with 4′,6-diamidino-2-phenylindole (DAPI, 1 μg mL$^{-1}$) and Wheat Germ Agglutinin (WGA conjugated to Alexa Fluor™ 488, 5 μg mL$^{-1}$, binding to chitinous cell walls, Thermo Fisher Scientific W11261)[104]. Bacteria were counted under a fluorescence microscope (Leitz Leica DMRB) at x1000 magnification. For each group and replicate, 20 *Synedra* cells (for attached bacteria) or 20 counting grids (for free-living bacteria) were examined, to reach representative mean values (standard error ≤5%).

Transparent Exopolymer Particles (TEP) and Coomassie Blue Stainable Particles (CSP) were determined microscopically and spectrophotometrically[54–56]. Volumes of 22 mL for spectrophotometric and 2 mL for microscopic analyses were gently filtered (<150 mbar) onto PC filters (0.4 μm, 25 mm, Nucleopore, Whatman). TEP filters were stained with Alcian Blue (AB, 0.02%, pH = 2.5) for 5 s, and CSP filters with Coomassie Brilliant Blue G (CBB-G, 0.04%, pH = 7.4) for 30 s. After staining, filters were rinsed with MilliQ to remove excess dye and stored at −20 °C. Sterile-filtered water (0.2 μm) from the cultures was used as blank. For spectrophotometric analyses, AB was extracted in 80% H$_2$SO$_4$ and CBB-G in 3% SDS in 50% isopropyl alcohol, as specified in Supplementary Methods S1. TEP concentrations are reported relative to xanthan gum (μg XG equivalents L$^{-1}$) and CSP concentrations relative to bovine serum albumin (μg BSA equivalents L$^{-1}$).

For microscopic analyses, the stained filters were placed onto CytoClear slides and inspected under a microscope (brightfield, x200 magnification). For each filter, 40 images were taken randomly along two transects across the entire filter area. Care was taken to use the same microscope and camera settings during the entire image acquisition. Image analyses were done in ImageJ 1.51p[105]. Images were corrected for uneven light intensities in the background and split into separate RGB channels. The blue channel was subtracted from the red channel, to remove chlorophyll-related areas and outlines of diatoms cells. The obtained 8-bit greyscale images were reversed in color and a global threshold was applied. Automated particle analyses in ImageJ provided the number of particles per filter area (used to calculate the number of particles per filtered volume, # mL$^{-1}$) and the cross-sectional area $A$ of each particle (used to calculate the equivalent circular diameter $ECD$, assuming spherical geometry, denoted as diameter). Particles with $ECD < 3$ μm were not included. Since the automatic TEP and CSP recognition was not perfect, the processed images were compared visually with the original images, and falsely outlined TEP or CSP were manually removed from the data set. Cell outlines and the infection stages were poorly visible on our images, and thus, we removed all cell-associated TEP and CSP outlines to not introduce any bias. Only freely-suspended TEP and CSP were therefore included in the microscopy analyses, whereas the spectrophotometric analyses included freely-suspended and cell-associated TEP and CSP.

To analyze the mass density of diatom cells (on day 6), 10 mL were transferred into 15 mL Falcon tubes and centrifuged (3000 rpm, 5 min). The supernatant was gently removed, and cells were re-suspended in 1 mL CHU-10 medium and transferred onto a density gradient, consisting of six viscous layers with increasing density from top to bottom (1.18–1.38 g cm$^{-3}$). The viscous layers were prepared with mixtures of Ludox TM colloidal silica (50 wt. % suspension in H$_2$O, Merck

420778), sucrose, and distilled water[49]. After 12 h and an additional centrifugation run (3000 rpm, 5 min), cells had settled into the density layer that was equivalent to the mass density $\rho_{s\text{-}cells}$ of their particulate material when hydrated in liquid (also termed solid hydrated density)[57]. The intact density layers (visible to the naked eye) were sub-sampled into 2 mL tubes and stored at 4 °C. Half of each sample was analyzed for its mass density $\rho_{s\text{-}cells}$ (g cm$^{-3}$) using a density meter (Anton Paar DMA 38). The other half was used to enumerate non-infected and infected diatom cells (after CFW staining) under an inverted fluorescence microscope with combined bright-field and UV excitation (x400 magnification, Olympus CKX41, Japan).

**Formation and characterization of sinking aggregates (model system, cylinder rotation).** Triplicates of non-infected and fungal-infected cultures were grown for six days until the infection prevalence had reached 59 ± 3% ($N = 3$ flasks) in the infected treatment. Thereafter, the cultures were diluted with CHU-10 medium to approx. 5000 cells mL$^{-1}$ (resembling cell abundances during bloom scenarios[87]) and transferred into rotating cylinders ($r = 8.5$ cm, $h = 10$ cm, $V = 2.3$ L, bubble-free) to mimic the aggregation of cells due to their stickiness and differential settling behavior (setup is shown in Supplementary Fig. S2). This technique allows forming aggregates that are commonly larger and sink faster than natural aggregates (e.g., compare with[106]) but their physical properties such as density, porosity, and composition are comparable to natural aggregates[107,108]. Initial diatom abundances were similar in both treatments (5363 ± 495 cells mL$^{-1}$ and 4765 ± 367 cells mL$^{-1}$, respectively, $P = 0.17$, t-test, $N = 3$ cylinders). Cylinders were rotated at 1–2 rpm at 17 °C in darkness on a roller table. The aggregate formation was recorded over time using a Mini Deep-focus Particle Imager (MDPI, Bellamare, La Jolla, CA, USA). The MDPI is a 'shadowgraph imager' using a near-infrared LED light source (A007 Indus star, LED dynamics, Randolph, VT, USA) behind a pinhole and a set of identical plano-convex collimator lenses to create parallel light beams between the LED light pod and the camera pod (Supplementary Fig. S2). Using this parallel light, $i.e.$, telecentric optics, ensured that all aggregates were in focus, independent of their position between the lenses[109]. We recorded two image sequences per cylinder every 2–9 hours. Each sequence lasted for at least one rotation (50 images). The sampling volume of two image sequences covered 2 × 0.35 L, equal to 30% of the entire cylinder volume.

Image sequences were processed similarly to TEP and CSP images (see above). Aggregates were counted, measured, and binned into size classes, in which the upper diameter was 1.3-times the lower size class. The resulting eleven size classes ranged from 0.4 to 5.6 mm (aggregates with an $ECD < 0.4$ mm were excluded from the data set, and aggregates with $ECD > 5.6$ mm were not present). We counted the number of aggregates per size class, reported as aggregate number concentration $N(d)$ (# L$^{-1}$). To describe the number concentration as a function of size, the aggregate size spectrum $n(d)$ (# L$^{-1}$ mm$^{-1}$) was calculated as

$$n(d) = \frac{N(d)}{\triangle d} \qquad (1)$$

where $\Delta d$ (mm) is the width of each size class ($d$ equals $ECD$)[110]. The volume spectrum $nVd$ was calculated by normalizing the volume distribution to the aggregate size

$$nVd = n(d) \times V_{agg} \times \triangle d \qquad (2)$$

where $V_{agg}$ (mm$^3$) is the average volume of single aggregates in each size class[111]. $V_{agg}$ and $d$ (equal to $ECD$) were calculated from $A$ derived from the image analyses and assuming spherical geometry. After stopping the rotation, individual aggregates were picked with a wide-bore glass pipette for various analyses. To ensure that aggregate sizes were similar for non-infected and infected aggregates, aggregates were sampled after 30 h from the non-infected treatment and after 45 h from the non-infected treatment (see Supplementary Fig. S3 for details). For subsequent analyses, aggregates were grouped into five sets (set I–V, each with 4–12 replicate aggregates) since not all parameters could be measured on the same aggregate.

Set I was used to determining the size, porosity, excess density, settling velocity, and fractal dimension. Aggregates were imaged directly after sampling using a compact camera (Panasonic DMC-TZ22, 14 MP). The cross-sectional area $A$ was determined in ImageJ and used to calculate the $ECD$ and volume $V$, assuming spherical geometry. The porosity $\varphi$ was calculated as

$$\varphi = 1 - \left( \frac{\Delta \rho \times V}{\rho_{s\text{-}agg} \times V} \right) \qquad (3)$$

where $\Delta \rho$ is the excess density of aggregates and $\rho_{s\text{-}agg}$ is the mass density of the particulate material in the aggregates when hydrated in liquid (solid hydrated density)[57]. We measured the settling velocity $U$ of aggregates in a settling column ($d = 11$ cm, $h = 40$ cm). The column was filled with the same water as the rotating cylinders and kept at the same temperature (17 °C). It was double-walled, with a 2 cm interspace filled with water and a sealed top, except for a 2 cm inlet to introduce the aggregates. This setup minimized convective currents in the column during settling experiments[112]. Each aggregate was allowed to freely settle from the wide-bore pipette into the water. The settling was timed along 15 cm using a stopwatch. Excess densities $\Delta \rho$ of aggregates were calculated using Stokes' law,

which applies to ballasted phytoplankton aggregates[113]

$$\triangle\rho = \frac{18\,U\,\nu}{g\,d^2} \qquad (4)$$

where $\nu$ is the kinematic viscosity ($1.085 \times 10^{-2}$ cm$^2$ s$^{-1}$ at 17 °C) and $g$ the gravitational acceleration (981 cm s$^{-2}$). The drag coefficient $C_D$ and Reynolds number $Re$ derived from

$$C_D = \frac{24}{Re} + \frac{6}{1 + Re^{0.5}} + 0.4 \qquad (5)$$

$$Re = \frac{d \times U}{\nu} \qquad (6)$$

We determined the fractal structure of aggregates, using their three-dimensional fractal number $D_3$. $D_3$ was derived from the slope of $d$ and $U$ after log–log transformation (Supplementary Fig. S4b)[112].

$$U \sim d^{D_3 - 1} \qquad (7)$$

Set II was used for oxygen respiration and particulate organic carbon (POC) analyses. Single aggregates were transferred into 5.9 mL gas-tight Exetainer® vials with 0.2 μm-filtered water from the rotating cylinders. Oxygen concentrations were measured with a Clark-type oxygen microsensor (OX-500, Unisense A/S, Denmark) immediately after aggregates were added and after 24 h-incubations at 17 °C in darkness. During incubations, the Exetainer rotated to keep the aggregates freely suspended and to minimize diffusion-limited gas exchange at the aggregate surface[114]. Oxygen consumption rates were converted to carbon-specific (C-specific) respiration rates by assuming a respiratory quotient of 1.2 mol O$_2$ to 1 mol CO$_2$[115] and by normalization to the aggregate-specific POC contents. Shifts in oxygen concentrations due to, e.g., temperature changes, were tested in control vials with 0.2 μm-filtered water but without aggregates. The observed shift in oxygen concentration was significantly lower in the control vials than in the vials with aggregates ($2 \pm 1\%$ and $11 \pm 6\%$, $N = 6$ and 18 vials, respectively, $P < 0.001$). After incubations, aggregates were filtered individually onto pre-combusted GF/F filters (25 mm, Whatman) and stored at $-20$ °C for later POC analyses. After storage, filters were fumed over HCl and dried at 50 °C overnight. POC was analyzed after combustion using a carbon analyzer (Surface C-800, Eltra Analysers, 0.01 mg C standard, 2% precision). Carbon respiration rates are given in the units d$^{-1}$ and % m$^{-1}$ (Fig. 5g, h), which define the fractional carbon respiration within single aggregates per day and per meter settled, respectively. Respiration rates per meter settled, also referred to as remineralization length scale $L$[6], were calculated as

$$L = \frac{C - specific\ respiration\ rate\,(d^{-1})}{U\,(m\,d^{-1})} \qquad (8)$$

for aggregates with $d = 1$–5 mm, based on the size~settling velocity curve fit (Fig. 5f) and mean respiration rates per day (Fig. 5g).

Set III and IV served to determine the infection prevalence and bacterial abundance. Single aggregates and 1 mL of the ambient water were sampled and inspected under the microscope, as described above (see section Culture characteristics). In short, the infection prevalence in *Synedra* populations was analyzed in Lugol-preserved samples (x400 magnification), whereas abundances of *Synedra*-associated bacteria were analyzed in PFA-preserved samples (x1000 magnification). Aggregates were disaggregated via gentle shaking in 2 mL microcentrifuge tubes prior to filtration and/or counting, to be able to inspect individual *Synedra* cells under the microscope. The number of aggregate-associated bacteria (per aggregate) was calculated by multiplying the number of associated bacteria per *Synedra*-cell type with the abundance of each *Synedra*-cell type per aggregate. We assumed 20,000 *Synedra* cells per aggregate since we counted on average $22,843 \pm 34,443$ *Synedra* cells per aggregate with $d = 2.1 \pm 0.8$ mm ($N = 18$).

Set IV was used to derive the mass density of single aggregates $\rho_{s\text{-}agg}$. Single aggregates were gently transferred on top of a density gradient. The further procedure resembled the protocol used for $\rho_{s\text{-}cell}$, i.e., after centrifugation and settling, the layer containing the aggregate was sampled and analyzed for its mass density (see above, last paragraph in the section Culture characteristics).

**Bacterial abundance and infection prevalence of aggregates formed from field-collected plankton communities (natural systems, cylinder rotation).** A plankton community was sampled from Lake Stechlin (Northern Germany) during the spring bloom in April 2018. Cells were concentrated from 0–10 m using a hand-held plankton net (25 μm mesh size, Hydro-Bios), resuspended in 10 L of in situ surface water, and transferred into three rotating cylinders. Cylinders rotated at 7.5 °C (in situ temperature, 1.5 rpm) for 12 h in darkness until macroscopic aggregates ($d = 2$–4 mm) were formed. Individual aggregates were sampled with a wide-bore glass pipette, and sub-volumes of the ambient water were sampled with a plastic syringe. Ten replicates (10x aggregates and $10 \times 2$ mL ambient water) were sampled from each cylinder and split in half. One half was preserved for determining the infection prevalence on single phytoplankton taxa, and the other half for counting cell-associated bacteria. Sample preservation and microscopy analyses were done as described above (see section Culture characteristics).

**Statistics and reproducibility.** Pairwise comparisons between mean values were calculated with the Mann–Whitney test for non-normally distributed data, the Welsh-test for normally distributed data with non-equal variance, and the *t*-test for normally distributed data with equal variance (agricolae v. 1.3.1 in R). Normal distribution was tested using the Shapiro-test and data variance with the *F*-test. Statistical differences between multiple groups were determined with the Kruskal–Wallis test (if normal distribution was rejected, with Bonferroni correction for *P*-value adjustment). Statistically significant differences between time series data were calculated based on generalized linear mixed models (GLMM, lmerTest v. 3.1.3 in R). Data were log-transformed if the assumption of homogeneity of variance was rejected (inspected visually on residuals versus fits plots). Pair-wise comparisons in time series data were run with emmeans (v. 1.7.2 in R). Statistical tests and plotting were done in R 3.3.0[116] and Origin2021. Statistical tests were run two-sided. The significance level was 0.05. *P* values at very highly significant levels are given as <0.001 or <0.0001. Uncertainties ($\pm$sd) that derived from combined uncertainties of single variables were calculated following the laws of error propagation. The number of replicates is indicated for each mean value in the results section.

**Reporting summary.** Further information on research design is available in the Nature Portfolio Reporting Summary linked to this article.

## Data availability
Data plotted in the figures and mentioned in the text are listed in the supplementary data files (Supplementary Data 1–6). Images used for particle analyses and the image processing code can be downloaded on Dryad https://doi.org/10.5061/dryad.2rbnzs7s7. Further data requests should be directed to the corresponding author.

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

## Acknowledgements

We are grateful to Hannah Bachmann, Matthias Bodenlos, Monika Degebrodt, Hannah Gebhardt, Maren Lentz, Uta Mallok, Monika Papke, Solvig Pinnow, Ignacio Rodanes Ajamil, Michael Sachtleben, and Kensuke Seto for their fantastic support during experiments and sample analyses. We especially thank Christiane Hassenrück for statistical advice. Funding was provided by the German Science Foundation DFG grant IV 124/3-1 (to H.P.G. and M.H.I. to fund the project and support I.K.) and the Emmy Noether Project KL 3332/1-1 (to I.K. to fund the project and support I.K.). Further funding was provided by the DFG (GR1540-33-1 to H.P.G. to support the culture maintenance), the Helmholtz Association Young Investigator Group SeaPump VH-NG-1000 (to M.H.I. to support C.M.F. and M.H.I.), the AWI Strategy Fund project "Eco-Pump" (to C.M.F. to support C.M.F.), the Japan Society for the Promotion of Science (grants JSPS KAKENHI 15KK0026, 16H02943, and 19H05667 to M.K. to support M.K.), and the DFG-Research Center of Excellence "The Ocean Floor – Earth's Uncharted Interface": EXC-2077-390741603 and framework of the HGF Infrastructure Program FRAM of the Alfred-Wegener-Institute Helmholtz Center for Polar and Marine Research (to M.H.I. to support M.H.I.). The MDPI was financed by the German Federal Ministry of Education and Research, BMBF (033W034A to J.C.N.).

## Author contributions

Conceptualization: I.K. and S.Vd.W. Methodology: I.K., S.Vd.W, M.H.I., T.J.W.W., C.C.N., J.C.N., M.K., and H.P.G. Investigation: I.K., S.Vd.W., C.M.F., and M.K. Visualization: I.K. Supervision: I.K., M.H.I., and H.P.G. Writing original draft: IK. Proofreading and editing: I.K., S.Vd.W., M.H.I., T.J.W.W., C.M.F., C.C.N., J.C.N., M.K., and H.P.G.

## Funding

## Competing interests

The authors declare no competing interests.

**Additional information**

