## [Peer Review File · Communications Biology]

Reviewers' comments:

Reviewer #1 (Remarks to the Author):

Klawonn et al. demonstrate that fungi may play a key role in the aquatic carbon cycle by using a model pathosystem that includes a *Synedra* diatom and the fungal parasite *Zygophlyctis*. They find that fungal infection inhibits cell aggregation, slows aggregate settling velocity and increases bacteria colonization and carbon respiration on infected particles. Based on their findings, they hypothesize that microparasites can regulate carbon fluxes in diverse aquatic systems, potentially leading to enhanced remineralization and reduced sedimentation of particulate organic carbon. The work by Klawonn et al. has a great deal of relevance for the understanding of the microscopic elements and the driving forces that govern the strength and efficiency of carbon flux in aquatic systems. In my opinion, the majority of the claims made by the authors are supported with data, the analysis is thorough and the paper fits well within the scope of *Communications Biology*. This is the second time I'm reviewing this paper and will therefore directly refer only to the comments that were unaddressed from the previous submission.

General comment:

1) As I suggested in the last version of the paper, I think it will be appropriate if the authors deposit all the images included in their study in a public repository. (Images of the filters used for cell count, images of particles, etc). Additionally, some of the supplementary figures miss the raw data that was used to produce them (e.g Fig S1). Please add it to the supplementary dataset.

Specific comments:

Line 175-178 – The authors write: “During the 9-day growth period, total *Synedra* abundances increased from approximately 0.8×10^4 to 3×10^4 cells mL⁻¹ (Fig. 2A), with similar growth rates for non-infected and infected *Synedra* populations during exponential growth (day 0–4, 0.26 ± 0.02 and 0.26 ± 0.02 d⁻¹, respectively, $P=0.64$, t-test, $N=3$ incubation flasks, see also Supplementary Fig. S1).”

The growth rate of the infected *Synedra* population seems faster than the non-infected population (in Fig. S1), yet you claim that in your calculation they are the same. Can you please explain it? Moreover, the growth rate of the infected *Synedra* population that you reported in the previous version was 0.31 ± 0.02 . How did you calculate the revised value? And how is it different from the previous calculation? Please include in the method the new way you used to calculate the growth rate.

Reviewer #2 (Remarks to the Author):

This paper addresses a very cool topic about the biological drivers of sinking aggregate dynamics, which could provide great micro-scale insights for large-scale ecosystem model. Particularly, the authors focused on studying how fungal parasites influence on phytoplankton-derived organic matter. I like the approach in which the authors designed both laboratory single cell culture experiments (also referred to as “model experiment”) and field sample experiments to confirm their hypotheses about the impacts of fungal parasites on particle dynamics at different scales. In both experiments, they measured biological and physical changes with time of phytoplankton-derived organic matter, including bacterial colonization, aggregate sizes, volumes and abundance, carbon respiration rate, and aggregate sinking rates, besides chemical factors (exopolymers) that influences on aggregate formation. Even though the infection was only introduced to one type of diatoms by one type of fungal parasites (which does not represent the diversity of the system), I think the experiments were implemented thoughtfully and can be set as a good example for future investigation. I appreciate the amount of work that the research team has put into this study, and I think that has shined through the paper. To improve the paper, I want to suggest a few minor things below:

1. Introduction:

- Can the authors consider adding in the overarching functional roles of fungal communities on phytoplankton particles before narrowing down the research into just fungal parasites? I am curious about the abundance and diversity of fungal communities on particles as well as their roles in particle carbon dynamics and how it compares to the impacts of the fungal parasites on particles.
- Personally, I think about fungi as carbon degraders just as much as bacteria or archaea, or sometimes fungi can play a role as symbionts with phytoplankton. What do the authors think about the relative contribution of fungal parasites compared to the rest of fungal members in an ecological perspective?
- Any relative comparison or literature out there about viral shunt vs. fungal shunt and their roles in sinking organic matter?

2. Material and Methods:

- Does the lab culture experiment resemble sinking condition as in the water column? Does the infected phytoplankton experience diffusion and sinking during 9 days of incubation?
- Why do we have to induce formation of aggregates? I thought the idea is to measure how fungal infection influences on the rate of aggregate formation. Will this introduce bias?
- What is the influence of fungal parasites on the disaggregation of the sinking aggregates?

3. Results:

- Fig 2+3: Error bars aren't clear which box that they belong to
- Fig 4 C-H: Need to redefine non-infected vs infected treatment on the plot or in figure caption

Reviewer #3 (Remarks to the Author):

General comments

This is a very interesting manuscript addressing the understudied role of fungal parasites in aquatic ecosystems. Particularly with reference to their influence on an important biogeochemical process, carbon cycling. The manuscript compares a cultured model system with data from a lake environment, showing how their results are applicable in environmental settings.

The authors present compelling evidence that fungal parasites lead to enhanced carbon remineralisation in the upper ocean and a reduced carbon export to depth in aquatic systems. Greater consideration of this process in models of carbon cycling will increase our understanding of this important process.

The statistics used are justified and robust, and the methods presented in a way allowing reproduction of the work.

Specific comments

Introduction

- Line 70-71: when referring to freshwater and coastal the authors can just cite the references, there is no need for the e.g.
- Line 83: I would use "regard" rather than "vein" here
- Line 96: instead of e.g., here the authors could use for instance...
- Line 104: the authors have "but see 40" in superscript, I think a short sentence outlining the important points of this reference here would be good
- Line 107: e.g not needed
- Line 130: only epidemics of fungal infection or could this be generally true of fungal parasitism?

Results

- Line 136: I think it would be useful to refer to the species name here for the organisms (or sp. for *Synedra* if not known)
- Line 189: NO₃⁻/NO₂⁻ implies a ratio of these 2, whereas I think the authors mean the sum of nitrate and nitrite?
- Line 208-213: Is the microscopically derived data shown anywhere? I cannot find it in the main text or supplementary.
- Figure 2: the authors use A/a and A/b and do not refer to this in the figure legend. They should either do this, or just change to be A) and B) and update the others to C, D and E.
- Line 235: I would use the term "cell stage/cell infection stage" rather than "cell type" as this could also refer to something else such as natural morphological differences
- Line 236-238: the authors here state the non-infected culture was colonised by fewer bacteria; however, the numbers and standard deviations do not appear to be statistically different.
- Figure 3: information for the plot 3A should be given in the figure legend here (I think this is provided in the methods section later, but would also be useful here)
- Line 262-265: here the authors give a comparison, and I was not sure if it was between the control and infected treatments, or the early and later samples of the infected treatment.
- Line 333-335: the discussion on respiration rates is something which should be moved to the discussion.
- Table 1: how did the authors define the different sets? Is this based on some criteria or was it based on the type of measurements they took?

Discussion

- Line 405: I would use multiple instead of multifarious (also line 417)
- Line 414: the e.g., is not needed here
- Line 433-434: were measurements of nutrients made in the lake also? There can be many different interactions between bacteria and phytoplankton aside from just nutrient depletion (see review by Buchan et al. 2014. Master recyclers: features and functions of bacteria associated with phytoplankton blooms, *Nature Reviews Microbiology*)
- Line 438-441: could there be bacterial-fungal interactions also, rather than related to phytoplankton which are important for aquatic systems?
- Line 446: no need for the "#" after 5
- Line 445: remove indeed: "TEP concentrations showed significantly..."
- Line 497-501: in virus cultures we also tend to see differences in the infection rates related to field populations. This is likely due to different host strains and their evolutionary relationship between host-virus but could be a consideration here also (for example see Short, 2012. The ecology of viruses that infect eukaryotic algae. *Environmental Microbiology*)
- Line 500-501: what was so challenging in the field samples vs. the culture system? It seems the definition of the different stages was quite well defined, so I am curious what was the challenge in the field (e.g., mixed communities)?
- Line 522-525: smaller aggregate size would also influence the interaction with the detritivore pool (e.g., small copepods)
- Figure 7: I find the bacteria on the figure to be quite hard to see, maybe you could increase the size of them slightly to make it more evident that you have an increase in bacterial abundance

Materials and methods

- Line 568-569: what does it mean by N=50 cells recently classified as *Ulnaria*?
- Line 616: what concentration of Lugol was used to fix the diatoms?
- Line 623: define excitation and emission the first times you use Ex and Em
- Line 620-622: if there were any strict criteria needed to be defined as a specific infection stage, here you could describe it. For instance, was there a certain size or feature to be considered a mature sporangium?
- Line 676: did the additional centrifugation step not interfere with the viscous layers which had been prepared?
- Line 690-693: did you fill the tanks to the top or was there gas space allowing interaction with air?
- Line 780: why is the remineralisation length scale equation not included as equation 8 and instead in the text?
- Line 795-797: why did the authors use this number since it sounds like they counted all the aggregates, or did they count a sub sample and then calculate the average and use this?

Reviewer #1

Klawonn et al. demonstrate that fungi may play a key role in the aquatic carbon cycle by using a model pathosystem that includes a *Synedra* diatom and the fungal parasite *Zygomycetis*. They find that fungal infection inhibits cell aggregation, slows aggregate settling velocity and increases bacteria colonization and carbon respiration on infected particles. Based on their findings, they hypothesize that microparasites can regulate carbon fluxes in diverse aquatic systems, potentially leading to enhanced remineralization and reduced sedimentation of particulate organic carbon. The work by Klawonn et al. has a great deal of relevance for the understanding of the microscopic elements and the driving forces that govern the strength and efficiency of carbon flux in aquatic systems. In my opinion, the majority of the claims made by the authors are supported with data, the analysis is thorough and the paper fits well within the scope of *Communications Biology*. This is the second time I'm reviewing this paper and will therefore directly refer only to the comments that were unaddressed from the previous submission.

We wish to thank the reviewer for the positive remarks and valuable suggestions to improve the manuscript. Thank you also for the previous review, which was highly constructive and thorough. In the following, we are pleased to provide a point-by-point reply. The reviewer's comments are indicated in bold, and ours in normal font.

Line numbers correspond to those in the "Revised Manuscript_Marked_Up" file (not the unmarked Article_file).

General comment:

1) As I suggested in the last version of the paper, I think it will be appropriate if the authors deposit all the images included in their study in a public repository. (Images of the filters used for cell count, images of particles, etc). Additionally, some of the supplementary figures miss the raw data that was used to produce them (e.g Fig S1). Please add it to the supplementary dataset.

Yes, we deposited the images at <https://datadryad.org> (release upon acceptance), as indicated in the data availability statement on pg. 35. We previously had indicated *figshare* as a repository, but now realized that Dryad allows larger datasets for upload. The uploaded images will include the particle images but no images for cell counts, unfortunately. We counted cells directly under the microscope (not on images) and thus no images are available from this analysis.

We added the data from Supplementary Figure S1 to the Supplementary Data.

Specific comments:

Line 175-178 – The authors write: “During the 9-day growth period, total *Synedra* abundances increased from approximately 0.8×10^4 to 3×10^4 cells mL⁻¹ (Fig. 2A), with similar growth rates for non-infected and infected *Synedra* populations during exponential growth (day 0–4, 0.26 ± 0.02 and 0.26 ± 0.02 d⁻¹, respectively, $P=0.64$, t-test, $N=3$ incubation flasks, see also Supplementary Fig. S1).”

The growth rate of the infected *Synedra* population seems faster than the non-infected population (in Fig. S1), yet you claim that in your calculation they are the same. Can you please explain it? Moreover, the growth rate of the infected *Synedra* population that you reported in the previous version was 0.31 ± 0.02 . How did you calculate the revised value? And how is it different from the previous calculation? Please include in the method the new way you used to calculate the growth rate.

Growth rates were calculated based on *Synedra* abundances on day 0 and day 4 (exponential growth phase with the steepest slope). Abundances were as follows (two treatments with three replicates each):

	Day 0 (=0.0 d) abundance cells_mL-1	Day 4 (=3.81 d) abundance cells_mL-1	growth rate d-1			
noninf treatment repl 1	7,773	19,685	0.244			
repl 2	7,770	19,849	0.246	mean	s.d.	
repl 3	7,025	20,175	0.277	0.26	0.02	
inf treatment repl 1	9,039	22,544	0.240			
repl 2	8,311	24,886	0.288	mean	s.d.	t.test
repl 3	8,155	22,422	0.265	0.26	0.02	P=0.64

These data are plotted in Supplementary Figure S1.

In the supplementary (*Further explanation to Supplementary Figure S1*), we explain the calculation and also give a potential explanation why growth rates may look faster.

*“The initial abundances of healthy, noninfected *Synedra* cells on day 0 were similar in both culture treatments (7,460±460 and 7,497±205 cells mL⁻¹ in the noninfected and infected culture, respectively, P=0.91). The higher total *Synedra* abundances in the fungal-infected culture in comparison to the noninfected culture on day 0 are explained by the added inoculum of infected cells. Those cells did not contribute to any further cell division since fungal infections are lethal to the diatom host cell. Growth rates μ of *Synedra* populations were calculated as*

$$\mu = \frac{\ln(N_{day\ 4}) - \ln(N_{day\ 0})}{(t_{day\ 4} - t_{day\ 0})} \quad (\text{Eq. S1})$$

where $N_{day\ 4}$ and $N_{day\ 0}$ are the cell abundances (including all cell types: noninfected, infected, and decaying cells) at $t_{day\ 4} = 0$ and $t_{day\ 0} = 3.8$ d.”

Our calculation in the previous version was incorrect since we used some wrong abundances as input data. Fortunately, your previous comment made us double-check the calculation and realize this error.

Reviewer #2

This paper addresses a very cool topic about the biological drivers of sinking aggregate dynamics, which could provide great micro-scale insights for large-scale ecosystem model. Particularly, the authors focused on studying how fungal parasites influence on phytoplankton-derived organic matter. I like the approach in which the authors designed both laboratory single cell culture experiments (also referred to as “model experiment”) and field sample experiments to confirm their hypotheses about the impacts of fungal parasites on particle dynamics at different scales. In both experiments, they measured biological and physical changes with time of phytoplankton-derived organic matter, including bacterial colonization, aggregate sizes, volumes and abundance, carbon respiration rate, and aggregate sinking rates, besides chemical factors (exopolymers) that influences on aggregate formation. Even though the infection was only introduced to one type of diatoms by one type of fungal parasites (which does not represent the diversity of the system), I think the experiments were implemented thoughtfully and can be set as a good example for future investigation. I appreciate the amount of work that the research team has put into this study, and I think that has shined through the paper. To improve the paper, I want to suggest a few minor things below:

#1. We wish to thank the reviewer for the positive review and valuable recommendations to improve the quality of the manuscript. We are pleased to provide a point-by-point reply in the following. The reviewer’s comments are indicated in bold, and ours in normal font.

Line numbers correspond to those in the “Revised Manuscript_Marked_Up” file (not the un-marked Article_file).

1. Introduction:

Can the authors consider adding in the overarching functional roles of fungal communities on phytoplankton particles before narrowing down the research into just fungal parasites? I am curious about the abundance and diversity of fungal communities on particles as well as their roles in particle carbon dynamics and how it compares to the impacts of the fungal parasites on particles. Personally, I think about fungi as carbon degraders just as much as bacteria or archaea, or sometimes fungi can play a role as symbionts with phytoplankton. What do the authors think about the relative contribution of fungal parasites compared to the rest of fungal members in an ecological perspective? Any relative comparison or literature out there about viral shunt vs. fungal shunt and their roles in sinking organic matter?

#2. We very much agree with the reviewer on those interesting aspects related to fungi. We do, however, favor to not broadening the introduction any further (simply to keep it as streamlined as possible). We believe that the functional role of parasitic fungi (studied by us) and saprotrophic fungi is different. Parasitic fungi seem comparable to viruses, as both infect live phytoplankton and utilize fresh organic matter. Saprotrophic fungi, by contrast, seem more comparable to saprotrophic bacteria, which are both capable of (but not limited to) utilizing aged detritus. In a recent publication, we directly compared the quantitative carbon transfer through the fungal vs. viral shunt (Klawonn et al 2021) (cited in l. 121). A similar comparison but in relation to sinking organic matter is, to our knowledge, not available yet.

2. Material and Methods:

Does the lab culture experiment resemble sinking condition as in the water column?

#3. We incubated the cultures for ca. 2 days in rotating cylinders which resembled sinking conditions. This technique has been introduced by Shanks and Edmondson (1989), and since then, become the standard methodology to form sinking phytoplankton aggregates in

the laboratory (e.g., Bhaskar et al 2005, Cruz and Neuer 2019, Grossart and Ploug 2001, Iversen et al 2010, Jackson 1994, Prairie et al 2015, Riebesell et al 1991). The setup can be seen as a laboratory coagulation reactor, in which solid body motion is established shortly after starting the rotation. Cells and particles then coagulate and form growing particles if they encounter each other due to their differential settling and if they are sufficiently sticky. Thus, the differential settling behavior of cells and particles is the major coagulation control in roller tanks (Jackson 1994). The same coagulation principle is also effective in nature. In nature, however, aggregation is more complex (as in any other laboratory vs. field comparison) since water turbulence is present and plankton communities are more diverse in, e.g., cell size, shape, function, and taxonomy. Aggregates formed in rotating cylinders are thus often larger than natural aggregates but their density, porosity, and composition are comparable (Shanks and Edmondson 1989). A recent re-evaluation of rolling tanks as “coagulation reactors” is available from (Jackson 2015), who confirmed the applicability of rotating tanks for studying coagulation processes.

In the Material and Method section, we briefly include this information. For instance, we include some of the references and mention that “[...] *to mimic the aggregation of cells due to their stickiness and differential settling behavior (setup is shown in Supplementary Fig. S2). This technique allows forming aggregates that are commonly larger and sink faster than natural aggregates (Jackson 2015), e.g., compare with (Alldredge and Gotschalk 1988), but their physical properties such as density, porosity, and composition are comparable to natural aggregates (Shanks and Edmondson 1989).*” (l. 721)

Does the infected phytoplankton experience diffusion and sinking during 9 days of incubation?

#4. The 9-day incubations were done in 1 L Erlenmeyer flasks which were loosely covered with aluminum foil. Thus, air exchange (incl. diffusion) was possible. Cells also sank within the flasks, but we resuspended the cells once per day through manual gentle shaking. We have now added the shaking information in l. 618.

Note that we used two sets of incubations, one lasting for 9 days and the other one for 6 days. In the M&M section, this is specified as:

“To monitor the culture development over time, 6x 650 mL (three per treatment) were grown and sub-sampled for nine days (see culture characteristics). The remaining 6x 650 mL (three per treatment) were grown for six days and subsequently transferred into rotating cylinders, to follow the formation of aggregates (see formation and characterization of sinking aggregates). Erlenmeyer flasks were gently shaken by hand once per day for cell resuspension and water mixing.” (l. 613)

For clarity, we have also specified these two sets of incubations at the beginning of the Results section, and we are now referring to each set in the headings throughout the Results section.

“One set of both treatments (each in triplicates) was grown and sub-sampled over a growth period of nine days, to monitor the culture development (referred to as culturing). An additional set was grown for six days and thereafter transferred into rotating cylinders, to analyze the formation and characteristics of aggregates (referred to as rotating cylinders).” (l. 154)

Why do we have to induce formation of aggregates? I thought the idea is to measure how fungal infection influences on the rate of aggregate formation. Will this introduce bias?

#5. Please see above (reply #3). We have now replaced “induce” with “follow” and “mimic” to avoid misunderstandings (l. 616 and l. 721).

What is the influence of fungal parasites on the disaggregation of the sinking aggregates?

#6. This would be very interesting to investigate. We speculate, based on our data, that fungal parasites facilitate particle disaggregation since particles were less abundant and smaller during the presence of the parasite. Final conclusions on this aspect, however, should await further investigations.

3. Results:

Fig 2+3: Error bars aren't clear which box that they belong to

#7. Good point. We now resolved this issue by (1) plotting only the plus direction of each error bar and (2) using different colors for each error bar. We applied the same fix also to Figure 3B, 4A, and 4D. If uncertainties remain, we refer the reader to Supplementary Data 1–3, which include the detailed numbers plotted in each figure.

Figure 2. Characteristics of the diatom co-culture in the absence or presence of the fungal microparasite. (A) Diatom abundances and infection prevalence in the non-infected (A/a) and fungal-infected culture (A/b), (B) chlorophyll a, (C) TEP concentrations, and (D) CSP concentrations, monitored over nine days. Data in (A) are given as mean±SD. To avoid overlapping of error bars, only the plus direction is shown and symbols for prevalence in (A/b) are offset to the left (by 0.3 days). Data in (B–D) are shown as single data points, while lines connect mean values over time (symbols for mean values are not displayed). P -values (shown in the upper right corner of each graph) indicate statistically significant differences between the entire time series of each parameter in both treatments (GLMM). Statistically significant differences between treatments at single time points are indicated with asterisks (* $P<0.05$, *** $P<0.001$). (A–D) $N=3$ incubation flasks.

Fig 4 C-H: Need to redefine non-infected vs infected treatment on the plot or in figure caption

#8. We added “upper panel” and “lower panel” to refer to the non-infected and infected treatment, respectively. For clarity, we also adjusted the positioning of the labels “Non-infected treatment” and “Fungal-infected treatment” in the figure.

References

- Allredge, A. L., Gotschalk, C. (1988). In situ settling behavior of marine snow. *Limnol. Oceanogr* **33**: 351.
- Bhaskar, P. V., Grossart, H. P., Bhosle, N. B., Simon, M. (2005). Production of macroaggregates from dissolved exopolymeric substances (EPS) of bacterial and diatom origin. *FEMS Microbiol. Ecol.* **53**: 255-264.
- Cruz, B. N., Neuer, S. (2019). Heterotrophic bacteria enhance the aggregation of the marine picocyanobacteria *Prochlorococcus* and *Synechococcus*. *Front. Microbiol.* **10**.
- Grossart, H. P., Ploug, H. (2001). Microbial degradation of organic carbon and nitrogen on diatom aggregates. *Limnol. Oceanogr.* **46**: 267-277.
- Iversen, M. H., Nowald, N., Ploug, H., Jackson, G. A., Fischer, G. (2010). High resolution profiles of vertical particulate organic matter export off Cape Blanc, Mauritania: Degradation processes and ballasting effects. *Deep-Sea Res. Pt I* **57**: 771-784.
- Jackson, G. A. (1994). Particle trajectories in a rotating cylinder: implications for aggregation incubations. *Deep-Sea Res. Pt I* **41**: 429-437.
- Jackson, G. A. (2015). Coagulation in a rotating cylinder. *Limnol. Oceanogr. Methods* **13**: e10018.
- Klawonn, I., Van den Wyngaert, S., Parada, A. E., Arandia-Gorostidi, N., Whitehouse, M. J., Grossart, H.-P., Dekas, A. E. (2021). Characterizing the “fungal shunt”: Parasitic fungi on diatoms affect carbon flow and bacterial communities in aquatic microbial food webs. *Proc. Natl. Acad. Sci. USA* **118**: e2102225118.
- Prairie, J. C., Ziervogel, K., Camassa, R., McLaughlin, R. M., White, B. L., Dewald, C., Arnosti, C. (2015). Delayed settling of marine snow: Effects of density gradient and particle properties and implications for carbon cycling. *Mar. Chem.* **175**: 28-38.
- Riebesell, U., Schloss, I., Smetacek, V. (1991). Aggregation of algae released from melting sea ice: implications for seeding and sedimentation. *Polar Biol.* **11**: 239-248.
- Shanks, A. L., Edmondson, E. W. (1989). Laboratory-made artificial marine snow: a biological model of the real thing. *Mar. Biol.* **101**: 463-470.

Reviewer #3

General comments

This is a very interesting manuscript addressing the understudied role of fungal parasites in aquatic ecosystems. Particularly with reference to their influence on an important biogeochemical process, carbon cycling. The manuscript compares a cultured model system with data from a lake environment, showing how their results are applicable in environmental settings. The authors present compelling evidence that fungal parasites lead to enhanced carbon remineralisation in the upper ocean and a reduced carbon export to depth in aquatic systems. Greater consideration of this process in models of carbon cycling will increase our understanding of this important process. The statistics used are justified and robust, and the methods presented in a way allowing reproduction of the work.

We wish to thank the reviewer for this thorough review, which helped us improve the manuscript. In the following, we are pleased to provide a point-by-point reply. The reviewer's comments are indicated in bold, and ours in normal font.

Line numbers correspond to those in the "Revised Manuscript_Marked_Up" file (not the un-marked Article_file).

Specific comments

Introduction

Line 70-71: when referring to freshwater and coastal the authors can just cite the references, there is no need for the e.g.

Revised as suggested. (l. 78/79)

Line 83: I would use "regard" rather than "vein" here

Revised as suggested. (l. 91)

Line 96: instead of e.g., here the authors could use for instance...

Revised as suggested. (l. 104)

Line 104: the authors have "but see 40" in superscript, I think a short sentence outlining the important points of this reference here would be good

We agree and integrated the references better into the sentence.

Chytrid infections thus occur frequently in freshwater and coastal regions, where a strong benthic–pelagic coupling is commonly assumed (Griffiths et al 2017, Renaud et al 2008), whereas in open ocean regions, their distribution is less evident with rare observations, so far (Lepère et al 2016). (l. 110)

Line 107: e.g not needed

Revised as suggested. (l. 115)

Line 130: only epidemics of fungal infection or could this be generally true of fungal parasitism?

l. 138: Yes, we suggest so. In a recent study, it was shown that Chytridiomycota can be present throughout the year, but their peak abundances coincide with phytoplankton blooms, i.e., they become epidemic (Van den Wyngaert et al 2022). Thus, parasitic chytrids may have a continuous impact throughout the year, but during epidemics, their control mechanism on organic matter fluxes is most impactful.

Results

Line 136: I think it would be useful to refer to the species name here for the organisms (or sp. for *Synedra* if not known)

We agree. The organisms are now specified as *Synedra* sp. and *Zygophlyctis planktonica*. (l. 144/145)

Line 189: NO₃⁻/NO₂⁻ implies a ratio of these 2, whereas I think the authors mean the sum of nitrate and nitrite?

True. We revised the text as NO₃⁻ + NO₂⁻. (l. 198)

Line 208-213: Is the microscopically derived data shown anywhere? I cannot find it in the main text or supplementary.

These data are not plotted but detailed numbers are included in Supplementary Data 6.

Figure 2: the authors use A/a and A/b and do not refer to this in the figure legend. They should either do this, or just change to be A) and B) and update the others to C, D and E.

We now refer to A/a and A/b in the legend. (l. 228)

Line 235: I would use the term “cell stage/cell infection stage” rather than “cell type” as this could also refer to something else such as natural morphological differences

We agree and now use the term “cell stage”. (l. 247)

Line 236-238: the authors here state the non-infected culture was colonised by fewer bacteria; however, the numbers and standard deviations do not appear to be statistically different.

l. 248–245. Yes, the difference between bacterial abundances was statistically significant. The difference can also be seen in Figure 3A. To clarify, we here copy/pasted this figure.

Statistical differences were derived from multiple groups comparison (Kruskal–Wallis), while different letters (d vs. c and b vs. a) denote significantly different groups. As a confirmation, we now also ran a pair-wise comparison (Wilcoxon-test) to compare both cell stages, which provided the same conclusion ($P=2.748e-06$ and $P=0.01264$).

Figure 3: information for the plot 3A should be given in the figure legend here (I think this is provided in the methods section later, but would also be useful here)

Fig. 3A, 5A, 6A and 6B have the same plot properties which we define in the Material and Method section “Data in Fig. 3A, 5A, and 6A/B are shown as single data points (white circles), the 25th, 50th, and 75th percentile (grey boxes), mean values (color-filled circles), and distribution curves (kernel smoothing).” (l. 870). We are now referring to this information in the figure legend (l. 264). To avoid repetition, we chose to include the information in the Method section (Statistics and reproducibility) rather than in each legend.

Line 262-265: here the authors give a comparison, and I was not sure if it was between the control and infected treatments, or the early and later samples of the infected treatment.

We are here referring to the comparison between the non-infected (control) and infected treatment. “[...] aggregates were less abundant and smaller in fungal-infected vs. non-infected *Synedra* treatments. (l. 276)

Line 333-335: the discussion on respiration rates is something which should be moved to the discussion.

l. 350: In general, we would agree; however, the magnitude of respiration in comparison to previous measurements is not relevant for the discussion on non-infected vs. fungal-infected aggregates. We thus find it more logical to provide this information in the results section.

Table 1: how did the authors define the different sets? Is this based on some criteria or was it based on the type of measurements they took?

Sets were chosen based on two criteria: (1) having at least four replicates for each parameter, and (2) measuring as many parameters as possible on one set. In an ideal case scenario, we would have liked to measure all parameters on each aggregate. Unfortunately, this was not possible since some measurements made the aggregates unusable for any subsequent measurements (“Aggregates were grouped into five sets (set I–V, each with 4–12 replicates) since not all parameters could be measured on the same aggregate.”, l. 378).

Discussion

Line 405: I would use multiple instead of multifarious (also line 417)

We agree and followed this suggestion. (l. 424)

Line 414: the e.g., is not needed here

Revised as suggested. (l. 433)

Line 433-434: were measurements of nutrients made in the lake also? There can be many different interactions between bacteria and phytoplankton aside from just nutrient depletion (see review by Buchan et al. 2014. Master recyclers: features and functions of bacteria associated with phytoplankton blooms, Nature Reviews Microbiology)

We did not measure nutrients during our sampling campaign but nutrient data are available from a monitoring program conducted in the same sampling area and month. Based on those, nutrient concentrations are commonly rather low in the water surface (5 m) during April:

	Concentration in μM		
	mean	sd	N*
NO_3^-	1.6	0.7	9
NO_2^-	0.15	0.07	8
NH_4^+	2.4	1.2	11
SRP	0.12	0.15	11

*N denotes the number of sampling dates in April (from 2006–2016)

In the discussion, we include two aspects of bacteria–phytoplankton interactions— bacterial chemotaxis towards decomposing phytoplankton and chemical signaling to repel opportunistic bacteria. We feel like those two are the most likely explanations. However, we agree that bacteria–phytoplankton interactions go beyond those aspects. As a compromise, we now included the review by Buchan et al. as a reference. (l. 457)

Line 438-441: could there be bacterial-fungal interactions also, rather than related to phytoplankton which are important for aquatic systems?

Potentially yes. But in a previous study, we found that fungal parasites are rarely colonized by bacteria, and if bacteria were colonizing the parasite, those bacteria received low amounts of phytoplankton-derived carbon in comparison to bacteria that directly colonized the phytoplankton cell (Klawonn et al 2021). This hints at a rather antagonistic bacteria–parasite interaction, which may not explain the bacterial increase on infected diatom cells.

Line 446: no need for the “#” after 5

Revised as suggested. (l. 465)

Line 445: remove indeed: “TEP concentrations showed significantly...”

Revised as suggested. (l. 474)

Line 497-501: in virus cultures we also tend to see differences in the infection rates related to field populations. This is likely due to different host strains and their evolutionary relationship between host-virus but could be a consideration here also (for example see Short, 2012. The ecology of viruses that infect eukaryotic algae. Environmental Microbiology)

Yes, good point. We added this aspect to the discussion (l. 514). It would be exciting to follow up on this in future projects.

Line 500-501: what was so challenging in the field samples vs. the culture system? It seems the definition of the different stages was quite well defined, so I am curious what was the challenge in the field (e.g., mixed communities)?

The challenge was three-fold:

- 1) In culture, the fungal morphology resembles textbook illustrations since growth conditions for the phytoplankton host are ideal (nutrients, light, etc.) and host abundances are plentiful for the parasite. In contrast, in natural populations, the infection stages are morphologically more variable, partly deviating from the textbook. For instance, an encysted zoospore may look similar to a matured sporangium that remained small in size due to other co-occurring infections on the same host cell or due to insufficient metabolism of the phytoplankton host.
- 2) Infection prevalence was rather low during our sampling (2%), which made it difficult to get a good overview of the different infection stages, and
- 3) while 1) and 2) hold true, it is still possible to distinguish between infection stages in natural populations. However, by the time we analyzed the Lugol samples, we did not attempt to do so since by then we did not yet know that the infection stage has an impact on the aggregation potential.

Line 522-525: smaller aggregate size would also influence the interaction with the detritivore pool (e.g., small copepods)

True. We do not discuss this aspect specifically to keep the overall discussion streamlined, but we mention the link between parasitic infections and zooplankton, and the related unknowns, in l. 432.

“[...] parasitic infections induce cascading effects on phytoplankton community composition (Gsell et al 2013), zooplankton production (Agha et al 2016, Rasconi et al 2020), and carbon recycling (Klawonn et al 2021), all of which are known to alter vertical mass fluxes (Boyd and

Newton 1999, Cavan et al 2017, Guidi et al 2009, Iversen et al 2010, Laurenceau-Cornec et al 2015, Steinberg et al 2008, van der Jagt et al 2020). The weighted impact of parasite epidemics on sinking organic matter fluxes thus likely varies in multifarious plankton networks—variations that we are only starting to disentangle.”

Figure 7: I find the bacteria on the figure to be quite hard to see, maybe you could increase the size of them slightly to make it more evident that you have an increase in bacterial abundance

To make the bacteria more evident, we changed their color to brown and also increased the number of drawn bacteria.

Figure 7. Illustration of the impact of fungal parasites on biological and physical properties of single diatom cells (A) and aggregates (B).

Materials and methods

Line 568-569: what does it mean by N=50 cells recently classified as Ulnaria?

N=50 cells refers to the number of cells on which we measured the given dimensions.

Recently classified as Ulnaria indicates the taxonomic uncertainty related to the genus *Synedra*. The embedded reference Williams (2011) suggests using *Ulnaria* instead of *Synedra*. Both genera names are thus often used as synonyms while a final taxonomic classification seems undecided. We have now revised the sentence as “*The model pathosystem comprised the pennate diatom host Synedra sp. Ehrenberg, 1830 (also referred to as Ulnaria (Williams 2011), strain HS-SYN2, l=87.6±4.1, w=3.3±0.5, and h=5.0±0.8 μm, N=50 cells) ...*” (l. 589).

Line 616: what concentration of Lugol was used to fix the diatoms?

We added 50 μL Lugol per 100 mL sample. This information is now included (l. 641). The Lugol recipe is specified in Supplementary Table S1.

Line 623: define excitation and emission the first times you use Ex and Em

Ex and Em are now defined as excitation wavelength and emission wavelength. (l. 646)

Line 620-622: if there were any strict criteria needed to be defined as a specific infection stage, here you could describe it. For instance, was there a certain size or feature to be considered a mature sporangium?

We define the cell stages in the Results section since we wanted to provide this information as early as possible in the text (l. 163–170). The sizes of sporangia and zoospores are included in the M&M section (l. 591/592 and 594).

Line 676: did the additional centrifugation step not interfere with the viscous layers which had been prepared?

The viscous layers were undisturbed and still visible after centrifugation, which was confirmed by our density measurements of each layer after centrifugation and sub-sampling. We added this information to the text (“*The intact six density layers (visible to the naked eye) were sub-sampled into 2 mL Eppendorf tubes and stored at 4°C.*”). (l. 707)

Line 690-693: did you fill the tanks to the top or was there gas space allowing interaction with air?

The cylinders were filled without gas space, as now indicated by the term “bubble-free” (l. 720). This is the common procedure when following the aggregation potential within rotating cylinders, as a solid-body motion is desired within the cylinders. Otherwise, fragile aggregates would disaggregate when getting in contact with bubbles as bubbles cause unwanted turbulence and friction.

Line 780: why is the remineralisation length scale equation not included as equation 8 and instead in the text?

The equation is now included as Eq. 8 (l. 815).

Line 795-797: why did the authors use this number since it sounds like they counted all the aggregates, or did they count a sub sample and then calculate the average and use this?

For simplicity, we rounded down the counted 22,843 cells per aggregate to 20,000 (l. 830). Our main message—the 5-times higher bacterial abundances on infected aggregates in comparison to non-infected aggregates—was not impacted by this rounding.

References

Agha, R., Saebelfeld, M., Manthey, C., Rohrlack, T., Wolinska, J. (2016). Chytrid parasitism facilitates trophic transfer between bloom-forming cyanobacteria and zooplankton (*Daphnia*). *Sci Rep.* **6**: 35039.

Boyd, P. W., Newton, P. P. (1999). Does planktonic community structure determine downward particulate organic carbon flux in different oceanic provinces? *Deep-Sea Res. Pt I* **46**: 63-91.

Cavan, E. L., Henson, S. A., Belcher, A., Sanders, R. (2017). Role of zooplankton in determining the efficiency of the biological carbon pump. *Biogeosciences* **14**: 177-186.

Griffiths, J. R., Kadin, M., Nascimento, F. J. A., Tamelander, T., Törnroos, A., Bonaglia, S., Bonsdorff, E., Brüchert, V., Gårdmark, A., Järnström, M., Kotta, J., Lindegren, M., Nordström, M. C., Norkko, A., Olsson, J., Weigel, B., Žydelis, R., Blenckner, T., Niiranen, S., Winder, M. (2017). The importance of benthic–pelagic coupling for marine ecosystem functioning in a changing world. *Global Change Biol.* **23**: 2179-2196.

Gsell, A. S., De Senerpont Domis, L. N., Verhoeven, K. J. F., Van Donk, E., Ibelings, B. W. (2013). Chytrid epidemics may increase genetic diversity of a diatom spring-bloom. *The ISME Journal* **7**: 2057-2059.

Guidi, L., Stemmann, L., Jackson, G. A., Ibanez, F., Claustre, H., Legendre, L., Picheral, M., Gorskya, G. (2009). Effects of phytoplankton community on production, size, and export of large aggregates: A world-ocean analysis. *Limnol. Oceanogr.* **54**: 1951-1963.

Iversen, M. H., Nowald, N., Ploug, H., Jackson, G. A., Fischer, G. (2010). High resolution profiles of vertical particulate organic matter export off Cape Blanc, Mauritania: Degradation processes and ballasting effects. *Deep-Sea Res. Pt I* **57**: 771-784.

Klawonn, I., Van den Wyngaert, S., Parada, A. E., Arandia-Gorostidi, N., Whitehouse, M. J., Grossart, H.-P., Dekas, A. E. (2021). Characterizing the “fungal shunt”: Parasitic fungi on diatoms affect carbon flow and bacterial communities in aquatic microbial food webs. *Proc. Natl. Acad. Sci. USA* **118**: e2102225118.

Laurenceau-Cornec, E. C., Trull, T. W., Davies, D. M., De La Rocha, C. L., Blain, S. (2015). Phytoplankton morphology controls on marine snow sinking velocity. *Mar. Ecol. Prog. Ser.* **520**: 35-56.

Lepère, C., Ostrowski, M., Hartmann, M., Zubkov, M. V., Scanlan, D. J. (2016). In situ associations between marine photosynthetic picoeukaryotes and potential parasites – a role for fungi? *Environmental microbiology reports* **8**: 445-451.

Rasconi, S., Ptacnik, R., Danner, S., Van den Wyngaert, S., Rohrlack, T., Pilecky, M., Kainz, M. J. (2020). Parasitic chytrids upgrade and convey primary produced carbon during inedible algae proliferation. *Protist* **171**: 125768.

Renaud, P. E., Morata, N., Carroll, M. L., Denisenko, S. G., Reigstad, M. (2008). Pelagic–benthic coupling in the western Barents Sea: Processes and time scales. *Deep Sea Research Part II: Topical Studies in Oceanography* **55**: 2372-2380.

Steinberg, D. K., Van Mooy, B. A. S., Buesseler, K. O., Boyd, P. W., Kobari, T., Karl, D. M. (2008). Bacterial vs. zooplankton control of sinking particle flux in the ocean's twilight zone. *Limnol. Oceanogr.* **53**: 1327-1338.

Van den Wyngaert, S., Ganzert, L., Seto, K., Rojas-Jimenez, K., Agha, R., Berger, S. A., Woodhouse, J., Padisak, J., Wurzbacher, C., Kagami, M., Grossart, H.-P. (2022). Seasonality of parasitic and saprotrophic zoospore fungi: linking sequence data to ecological traits. *The ISME Journal*.

van der Jagt, H., Wiedmann, I., Hildebrandt, N., Niehoff, B., Iversen, M. H. (2020). Aggregate feeding by the copepods *Calanus* and *Pseudocalanus* controls carbon flux attenuation in the arctic shelf sea during the productive period. *Frontiers in Marine Science* **7**: 543124.

Williams, D. M. (2011). *Synedra*, *Ulnaria*: definitions and descriptions – a partial resolution. *Diatom Research* **26**: 149-153.

REVIEWERS' COMMENTS:

Reviewer #1 (Remarks to the Author):

I have no further comments to make.
I recommend this paper for publication

Reviewer #2 (Remarks to the Author):

I thank the authors for their very thorough responses to my review. I am satisfied with their responses and would like to suggest the editors to accept this article. Some of the information that could cause misunderstanding has been addressed in the Methods. Other missing pieces in the figures are also identified. I think this manuscript would add a lot to our understanding of the micro-scale dynamics and hopefully that will help us to develop a more detailed and robust large-scale ecosystem model about the fate of sinking organic matter.

Reviewer #3 (Remarks to the Author):

I would like to thank the authors for their thorough response to comments and edited version. I am happy with all of their responses and have no further comments.